 # TULIP: Token-length Upgraded CLIP

**Ivona Najdenkoska**[*]  **Mohammad Mahdi Derakhshani**[*]  **Yuki M. Asano**
**Nanne van Noord**  **Marcel Worring**[†]  **Cees G. M. Snoek**[†]
University of Amsterdam
Amsterdam, the Netherlands

## Abstract

We address the challenge of representing long captions in vision-language models, such as CLIP. By design these models are limited by fixed, absolute positional encodings, restricting inputs to a maximum of 77 tokens and hindering performance on tasks requiring longer descriptions. Although recent work has attempted to overcome this limit, their proposed approaches struggle to model token relationships over longer distances and simply extend to a fixed new token length. Instead, we propose a generalizable method, named TULIP, able to upgrade the token length to any length for CLIP-like models. We do so by improving the architecture with relative position encodings, followed by a training procedure that (i) distills the original CLIP text encoder into an encoder with relative position encodings and (ii) enhances the model for aligning longer captions with images. By effectively encoding captions longer than the default 77 tokens, our model outperforms baselines on cross-modal tasks such as retrieval and text-to-image generation. The code repository is available at https://github.com/ivonajdenkoska/tulip.

## 1 Introduction

Obtaining text representations for long captions in vision-language models is an open research challenge. Within the context of text-only large language models, this problem has been studied extensively (Chung et al., 2024; Touvron et al., 2023; Bai et al., 2023; Gu & Dao, 2023; Su et al., 2024; Dubey et al., 2024; Pawar et al., 2024; Jiang et al., 2024; Lieber et al., 2024), yet, methods for context length expansion have only scarcely made their way into the vision-language domain. For instance, contrastive vision-language models like CLIP (Radford et al., 2021; Jia et al., 2021; Li et al., 2022) are constrained to short input sequences, capped at 77 tokens. Natively, such models based on Transformers (Vaswani et al., 2017) process their input as an unordered set of tokens and use positional encoding to preserve order information (Yun et al., 2019). In particular, CLIP models use absolute positional encodings, which rely on a predefined maximum number of tokens, thereby limiting the input sequence to 77 tokens by design. We introduce an efficient and generalizable method called TULIP, that upgrades the context window size of CLIP-like models.

Recently, Long-CLIP (Zhang et al., 2024) highlighted this problem and proposed an approach for unlocking the long-text capabilities of CLIP. However, their approach focuses on stretching the existing absolute positional encodings, which primarily addresses the issue of the hard token limit but it continues to rely on absolute encodings which inhibit the model's comprehension of pairwise token relationships. To address the challenge of comprehensively modeling the pairwise distances between tokens, more flexible positional encoding methods have been proposed (Shaw et al., 2018; Su et al., 2024; Golovneva et al., 2024). For example, relative positional encodings offer an approach that allows the model to capture interactions between tokens more effectively, regardless of

---

[*]Shared first authorship. Corresponding authors: {i.najdenkoska, m.m.derakhshani}@uva.nl. The authors can change the order for their purposes.

[†]Shared last authorship; order random.

their placement in the sequence (Su et al., 2024). While this method has shown promise in natural language processing tasks, it remains unexplored in vision-language models. This is not surprising, as extending the context length in vision-language models by switching to more flexible positional encodings is computationally costly because of its significant retraining efforts.

Our proposed method upgrades the context window size of CLIP-like models by utilizing relative positional encodings, which are not restricted to a fixed length. Changing the model in this manner normally requires expensive multi-modal retraining. Instead, we propose an adaptation phase to first distill the knowledge of the original CLIP into the new model with relative positional encodings using only caption data. Such a distillation approach is flexible and can be applied to CLIP-like models constrained by the standard 77-token window, transforming it into a model capable of handling longer captions. Afterwards, we perform full fine-tuning on the distilled model for a single epoch on 1 million image-caption pairs (Chen et al., 2023b), to further improve the alignment between images and long captions. After these phases, our TULIP model can ingest captions longer than the usual 77 tokens and we observe improved performance on tasks ranging from cross-modal retrieval to text-to-image generation, compared to (interpolated) fixed token-window baselines.

In addition to our new method, we introduce a new benchmark for long captions adapted from the recently introduced Dense Captioning Images (DCI) dataset (Urbanek et al., 2024). This benchmark overcomes the limitations of existing retrieval benchmarks which lack diversity, as they focus on specific scenes (e.g., Urban-1K (Zhang et al., 2024)), or are in-distribution datasets with already saturated performance (e.g., ShareGPT4v test set (Chen et al., 2023b)). Our results show that evaluating in a true long caption setting is crucial for evaluating contrastive vision-language mode as they unearth an increased performance gap as compared to prior benchmarks.

In summary, our contributions are: (1) We propose TULIP, the first contrastive vision-language model with relative positional encodings for long captions. (2) We propose a general training procedure to adapt the positional encodings of CLIP-like models using a two-step adaptation encompassing relative position distillation and expansion. (3) We demonstrate improved performance across different cross-modal retrieval and image generation tasks. We define a new benchmark Long-DCI for a more comprehensive evaluation of long-caption retrieval tasks.

## 2 RELATED WORK

**Position Encodings in Transformer Models.** Transformers, the foundational architecture for many vision-language models, rely on positional encodings to compensate for the lack of inherent positional awareness in their set-based representation. In the natural language domain, absolute positional encodings (Vaswani et al., 2017) were initially proposed, where fixed embeddings are added to token embeddings based on their position in the sentence sequence. However, as models have grown larger and more complex, alternative approaches emerged such as relative positional encodings (Shaw et al., 2018; Press et al., 2021), randomized positional encodings (Ruoss et al., 2023), extrapolation techniques (Press et al., 2022) and positional interpolation (Chen et al., 2023c). Other works, such as Rotary Position Embedding (RoPE) (Su et al., 2024) and its variations (Chen et al., 2023c; Peng et al., 2023), apply relative positional encodings without any modifications to the self-attention mechanism, making it computationally efficient. Another recent approach is Contextual Position Encodings (CoPE) (Golovneva et al., 2024), which is a more general position encoding technique enabling one to attend to the $i$-th particular word, noun, or even sentence. In contrastive vision-language models, integrating positional information effectively across modalities remains an unsolved challenge and this is the primary focus of this paper.

**Contrastive Vision-Language Models with Long Captions.** Contrastive Vision-language models have made significant strides in aligning visual and textual modalities, driven by the success of large-scale pre-trained models such as CLIP (Radford et al., 2021), ALIGN (Jia et al., 2021), BLIP (Li et al., 2022) and many others (Garg et al., 2023; Vasu et al., 2024b;a; Cherti et al., 2023; Sun et al., 2023). These models leverage contrastive learning techniques to align image and text representations in a shared feature space, enabling robust performance on tasks such as image-text retrieval and zero-shot classification. However, all these models focus on short, global textual descriptions, often limited by small context windows, such as the 77-token limit in CLIP. Recent research has started considering this limitation, for instance, DCI (Urbanek et al., 2024) highlights the problem by pointing out that CLIP's 77-token limit restricts the model from accommodating detailed, dense

captions. Similarly, another work introduces DOCCI (Onoe et al., 2024), a captioning dataset that provides detailed image descriptions that capture key challenges such as spatial relations, counting, text rendering and world knowledge. DreamLIP (Zheng et al., 2024) studies the usage of synthetically generated long captions under the contrastive learning framework but uses subsampled short captions from longer captions instead of processing the complete long captions. Long-CLIP (Zhang et al., 2024) engages the challenge more directly, by stretching absolute position encodings through interpolation to enable fine-tuning with long captions. However, interpolation only partially addresses the limitations of absolute encodings when processing longer, complex captions. This is because interpolation merely extends the existing positional information without fundamentally altering its nature or capabilities. These limitations include diminished ability to capture fine-grained relative positions and poor generalization to longer captions (Pawar et al., 2024). We propose an alternative approach that incorporates relative encodings without training from scratch, thereby enabling more effective processing of long captions and the comprehension of the pairwise token relationships therein.

# 3 TULIP

In this section, we introduce our method Token-Length Upgraded CLIP (TULIP). We start by stating the problem setting, followed by introducing the positional encoding swapping and the two-step adaptation procedure: (i) relative position distillation and (ii) relative position expansion.

## 3.1 PROBLEM STATEMENT

Let model $f$ be a contrastive vision-language model, such as CLIP, designed to align text and images in a shared embedding space. The text encoder of $f$, denoted as $f_T$ is constrained to processing sequences up to a predefined number of 77 tokens, denoted by $T_f = 77$. This is due to the fixed absolute positional encodings $P_f \in \mathbb{R}^{77 \times d}$, where $d$ is the dimensionality of the embeddings. Given an input sequence $x = [x_1, x_2, \ldots, x_n]$, where $n > T_f$, the model truncates $x$ to $x' = [x_1, \ldots, x_{T_f}]$ losing critical information from the sequence beyond the first 77 tokens.

Our objective is to transform model $f$ into model $g$, without any re-training from scratch, to enable processing sequences of arbitrary length $T_g$. In this new model $g$, the token-length constraint of $f_T$ is removed, allowing the model to handle inputs of length $T_g > 77$.

## 3.2 POSITIONAL ENCODING SWAPPING

In $f_T$, the positional encoding function $P_f(i)$ maps each token position $i$ to a vector in $\mathbb{R}^d$, within a window of size 77. To overcome this limitation, we redefine the positional encoding function for model $g$, denoted as $P_g(i)$ which scales with input length $T_g$. This new function allows the model $g$ to process captions of arbitrary length without truncation.

We implement $P_g(i)$ as Rotary Positional encodings (RoPE) (Su et al., 2024) presented in Figure 1. Unlike traditional absolute positional encodings, where each position in a sequence is assigned a fixed vector, RoPE rotates the embeddings based on the relative distance between tokens. More specifically, we alter the calculation of the attention weights within the self-attention layers of the text encoder $f_T$, which are initially calculated as $\text{softmax}(\frac{q_m^T k_n}{\sqrt{d}})$, where:

$$q_m = W_q x_m, k_n = W_k x_n, \tag{1}$$

representing the query and key vectors respectively for the $m$-th and $n$-th tokens in the sequence $x$, and $W_q, W_k$ are their learned projection matrices. With RoPE, we inject the position information of each token $x_i$ into the $q_m$ and $k_n$ vectors by:

$$q_m = R_{\Theta,m} W_q x_m, \quad k_n = R_{\Theta,n} W_k x_n, \tag{2}$$

where $R_{\Theta,m}$ and $R_{\Theta,n}$ represent rotation matrices and $\Theta$ is the rotational frequency associated with the $j$-th dimension of the embedding. The rotational frequency ensures that different dimensions of the token embeddings rotate differently, embedding both absolute and relative positional information into the self-attention mechanism.

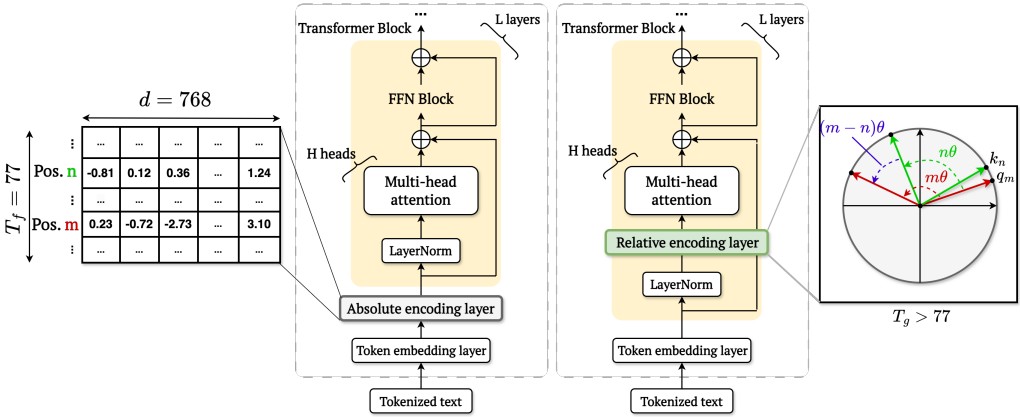

Figure 1: **Swapping the Positional Encoding.** We update CLIP models by replacing the absolute positional encoding with relative positional encoding in each transformer block. This modification allows for long caption understanding and better modeling of pairwise token dependencies.

### 3.3 RELATIVE POSITION DISTILLATION

Once the architectural changes are in place, the next challenge is to adapt the text encoder of model $g$ to handle both short and long text inputs while retaining the image-text alignment capabilities of model $f$. We achieve this through knowledge distillation, shown with the first block of Figure 2, where $f_T$ acts as the teacher and the text encoder with the new relative encodings as the student $s_T$. A key advantage of this approach is that it eliminates the need to retrain the model from scratch despite the introduction of new relative positional encodings. Instead, we distill the knowledge from $f_T$, transferring its capabilities to the student model without losing the original alignment performance. This method is not only efficient but also generalizable, making it applicable to any text encoder that requires adaptation to new positional encodings.

Let $x = [x_1, x_2, \ldots, x_n]$ represent an input caption, where $n \leq T_f$. Both the teacher model $f_T$ and the student model $s_T$ encode this sentence into a shared embedding space. Specifically, the teacher model $f_T$ aggregates the sequence into a special text token and yields the output embeddings $z_{f_T} = f_T(x)$. Similarly, the student model yields output embeddings $z_{s_T} = s_T(x)$. The distillation loss is formulated as a cosine similarity between $z_{f_T}$ and $z_{s_T}$, aiming to maximize their alignment:

$$\mathcal{L}_{\text{distill}} = \frac{z_{f_T} \cdot z_{s_T}}{\|z_{f_T}\|\|z_{s_T}\|}. \tag{3}$$

This loss function ensures that the student model $s_T$ learns from the teacher model $f_T$, by leveraging the original model's ability to align image and text. After the distillation, the student model $s_T$ has the capabilities of the teacher model for encoding up to 77 tokens with relative positional encodings. The only missing feature is the ability to process longer context, which is addressed next.

### 3.4 RELATIVE POSITION EXPANSION

Finally, we expand the context length of model $g$ beyond the original 77-token limit by fine-tuning it with longer captions, shown in the second block of Figure 2. We start by copying the student model weights into the text encoder of $g$. This is followed by employing Neural Tangent Kernel (NTK)-aware scaled RoPE (bloc97, 2023), a refined version that adapts to the changing length of the input. In particular, we aim to scale the rotational frequency $\Theta$ by a factor $(\alpha * \frac{T_g}{T_f}) - (\alpha - 1)$, where $\alpha$ is a hyperparameter, to accommodate the training of the new higher token positions. The objective behind such scaling is to resolve the problem of losing high-frequency information when interpolating the RoPE embeddings to the higher token positions. In model $g$, the new positional encodings now support input sentences $x_g$ of arbitrary length, $T_g > T_f$. We proceed with fine-tuning using the usual contrastive loss as follows: $\mathcal{L}(x, y) = -\log \frac{\exp(\cos(z_x, z_y)/\tau)}{\sum_{y'} \exp(\cos(z_x, z_{y'})/\tau)}$, where $x$ is the sentence, $y$ is the image and $\tau$ is the temperature hyperparameter. However, to keep the original

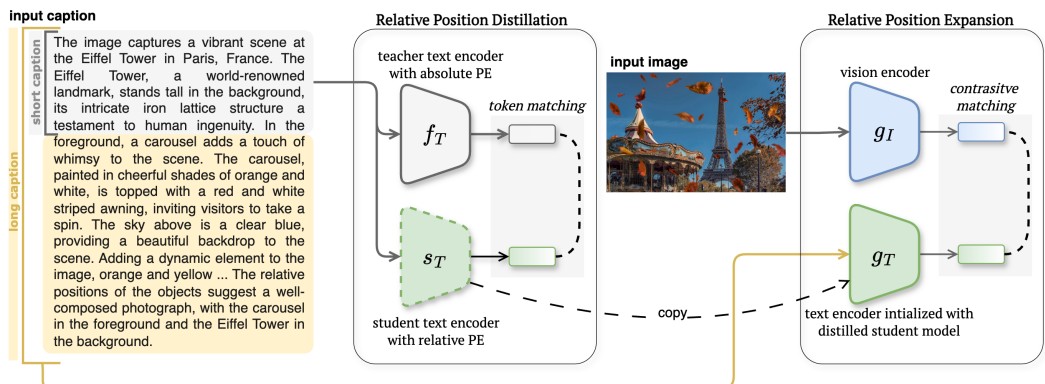

Figure 2: **TULIP training procedure.** First, we perform relative position adaptation by distilling the knowledge of the CLIP text encoder into a student text encoder initialized with relative position encodings. This stage uses the first 77 tokens of a long caption (the gray block). The second stage is the relative position expansion, where we fine-tune the distilled text encoder with captions longer than 77 tokens (the combined gray and yellow blocks), together with the vision encoder.

capability for handling short sentences as well, we jointly optimize two contrastive loss terms for both long and short sentences, as follows:

$$\mathcal{L}_{\text{total}}(x_{T_g}, x_{T_f}, y) = \lambda \times \mathcal{L}_{\text{short}}(x_{T_f}, y) + (1 - \lambda) \times \mathcal{L}_{\text{long}}(x_{T_g}, y). \quad (4)$$

The first loss term, $\mathcal{L}_{\text{short}}$, is designed to preserve the model's ability to handle short captions, aligning with the capabilities of model $f$. The other term, $\mathcal{L}_{\text{long}}$, is introduced to ensure that model $g$ can process longer captions effectively.

By extending the context length in this manner, model $g$ becomes capable of handling both short and long captions, making it more versatile for a wide range of vision-language tasks as we will demonstrate in the experiments.

## 4 EXPERIMENTS & RESULTS

**Datasets and downstream tasks.** We evaluate TULIP on three downstream tasks: short caption cross-modal retrieval, long caption cross-modal retrieval, and text-to-image generation. For short caption cross-modal retrieval, we follow Zhang et al. (2024) and evaluate our model on the COCO2017 5k validation set (Lin et al., 2014) and the full Flickr30k dataset (Plummer et al., 2015). Similarly, for long caption cross-modal retrieval, we use two datasets, namely ShareGPT4V test split and Urban-1K. Both datasets contain 1,000 image-caption pairs, collected by Zhang et al. (2024) and recaptioned using the ShareGPT4V captioning model.

|  |  | Long-DCI | | ShareGPT4V | | Urban-1K | |
|---|---|---|---|---|---|---|---|
|  |  | Img2Txt | Txt2Img | Img2Txt | Txt2Img | Img2Txt | Txt2Img |
| ViT-B-16 | CLIP | 35.9 | 33.7 | 78.2 | 79.6 | 68.1 | 53.6 |
|  | Fine-tuned CLIP | 46.3 | 45.4 | 94.1 | 93.6 | 80.4 | 79.8 |
|  | Long-CLIP | 42.1 | 48.4 | 94.6 | 93.3 | 78.9 | 79.5 |
|  | **TULIP (Ours)** | **50.2** | **50.6** | **98.6** | **98.6** | **88.1** | **86.6** |
| ViT-L-14 | CLIP | 35.0 | 37.0 | 81.8 | 84.0 | 68.7 | 52.8 |
|  | Fine-tuned CLIP | 51.6 | 50.7 | 95.3 | 95.4 | 78.0 | 76.5 |
|  | Long-CLIP | 54.0 | 46.1 | 95.8 | 95.6 | 82.7 | 86.1 |
|  | **TULIP (Ours)** | **55.7** | **56.4** | **99.0** | **99.0** | **90.1** | **91.1** |

Table 1: **Long caption cross-modal retrieval comparison** on Long-DCI, ShareGPT4V and Urban-1K. TULIP consistently outperforms other CLIP variants across all evaluated datasets and tasks. Note that we adopt the results for CLIP and Long-CLIP from Zhang et al. (2024), while we fine-tune CLIP (Fine-tuned CLIP) on ShareGPT4V ourselves.

| | | COCO | | | | Flickr30k | | | |
| | | Img2Txt | | Txt2Img | | Img2Txt | | Txt2Img | |
| | | R@1 | R@5 | R@1 | R@5 | R@1 | R@5 | R@1 | R@5 |
|---|---|---|---|---|---|---|---|---|---|
| ViT-B-16 | CLIP | 51.8 | 76.8 | 32.7 | 57.7 | 44.1 | 68.2 | 24.7 | 45.1 |
| | Fine-tuned CLIP | 37.4 | 62.3 | 21.8 | 43.4 | 25.7 | 45.8 | 17.9 | 34.5 |
| | Long-CLIP | **57.6** | **81.1** | 40.4 | 65.8 | **46.8** | **71.4** | 34.1 | 56.3 |
| | **TULIP (Ours)** | 56.8 | 80.3 | **40.7** | **66.1** | 46.1 | 70.8 | **35.2** | **57.4** |
| ViT-L-14 | CLIP | 56.1 | 79.5 | 35.4 | 60.1 | 48.5 | 72.6 | 28.0 | 49.3 |
| | Fine-tuned CLIP | 37.9 | 63.1 | 23.1 | 45.1 | 26.0 | 46.3 | 17.9 | 34.9 |
| | Long-CLIP | **62.8** | **85.1** | **46.3** | 70.8 | 53.4 | 77.5 | 41.2 | 64.1 |
| | **TULIP (Ours)** | 62.6 | 84.7 | 46.1 | **71.1** | **56.7** | **79.5** | **41.6** | **64.3** |

Table 2: **Short caption cross-modal retrieval comparison on COCO and Flickr30k.** TULIP shows competitive performance, often matching or exceeding Long-CLIP across different metrics and model backbones.

It is important to recognize that long-caption cross-modal retrieval benchmarks come with several limitations. These include a lack of diversity, as some focus on narrowly defined scenes (e.g., Urban-1K), while others consist of in-distribution datasets where performance is already saturated (e.g., ShareGPT4V). To overcome these challenges, we define a new benchmark for long captions adapted from the recently introduced Dense Captioning Images (DCI) dataset (Urbanek et al., 2024). **Long-DCI** includes 7,000 human-annotated images and *long* caption pairs, with captions averaging 200 tokens per image. This human-led annotation process ensures diverse and accurate descriptions, avoiding the biases inherent in AI-generated captions like those from ShareGPT4V.

**Evaluation metrics.** We report image-to-text and text-to-image retrieval performance using recall as the standard evaluation metric. Our evaluation process is the same for all experiments, including how we handle long input tokens. For each dataset, we choose the best design and settings using a validation set, and then report the final results on the test set.

**Training details.** Our training procedure comprises two phases: relative position distillation and relative position expansion, both utilize the ShareGPT4V dataset (Chen et al., 2023b) containing 1M image and long caption pairs. During the relative position distillation phase, we truncate captions to the first 77 tokens for both the teacher and student models. We train the student model using cosine loss as the distillation loss function for 20 epochs with a batch size of 640 using the AdamW optimizer (Loshchilov, 2017), setting the learning rate to 5e-4 with 1000 warmup steps. In the relative position expansion phase, we employ full-length captions without truncation, exposing the model to comprehensive-textual details. The full TULIP model, featuring the new distilled text encoder, is fine-tuned using the NTK approach, with $\alpha$ empirically set to 8.0. For all main experiments, we use 248 number of tokens to match Long-CLIP's context length for a fair comparison. Note that we can increase the length to more tokens, as shown in our ablations. We perform this fine-tuning stage for a single epoch with a batch size of 1280, a learning rate of 1e-5, and 1000 warmup steps using AdamW. We base our implementations on OpenAI's pre-trained CLIP-ViT-B-16 and CLIP-ViT-L-14 architectures (Ilharco et al., 2021).

## 4.1 CROSS-MODAL RETRIEVAL COMPARISON

We evaluate TULIP against the original CLIP, fine-tuned CLIP on ShareGPT4V (Chen et al., 2023b), and Long-CLIP (Zhang et al., 2024) for both long-caption (Table 1) and short-caption (Table 2) cross-modal retrieval tasks. As shown in Table 1, our proposed model outperforms all benchmarks in long-caption cross-modal retrieval across 3 datasets on both image-to-text and text-to-image retrieval, utilizing two different vision backbones: ViT-B-16 and ViT-L-14 (Dosovitskiy et al., 2021).

On the Long-DCI dataset, which is a more challenging benchmark for all approaches, we can observe that Long-CLIP already shows improvement over the original CLIP and that our TULIP model further improves on this. These results demonstrate the efficacy of TULIP in enhancing CLIP's capabilities for long-caption cross-modal retrieval, particularly in diverse scenarios. For short captions, as shown in Table 2, we find that the tailored approach used by Long-CLIP for the first 20 tokens is beneficial for the short caption performance as they outperform CLIP even on short captions. Whereas TULIP is able to obtain competitive performance without needing to specifically tailor to

the first 20 tokens, which demonstrates the flexibility of relative positional encodings across different caption lengths. Overall, the results in Table 1 and 2 indicate that TULIP is effective not only for long captions but also for maintaining competitive performance in short caption retrieval scenarios.

## 4.2 TEXT-TO-IMAGE GENERATION

In this section, we evaluate qualitatively how our method enhances text-to-image generation by simply replacing the original CLIP ViT-L-14 text encoder in Stable Diffusion XL (SDXL) (Podell et al., 2023) with TULIP. Note that we do not perform any additional training of the diffusion model. As observed in Figure 3, TULIP demonstrates improvements in both long and short caption understanding and modeling of nuanced details, compared to T5-based models such as PIXART-Alpha (Chen et al., 2023a) and ELLA (Hu et al., 2024), as well as CLIP-ViT-L-14 and Long-CLIP (Zhang et al., 2024). For example, our version of SDXL + TULIP accurately depicts *"the red tulip inside a wooden box"* in the first example. This is not the case with SDXL + CLIP or SDXL + Long-CLIP which generate many tulips in the garden, missing the key detail of a single tulip being inside the box. This observation shows that our model can indeed capture finer details in the description. The second example shows that our model successfully generates *"an old man sits on a rock"* even when this detail appears beyond the 77-token limit. On the other hand, base CLIP, Long-CLIP and PIXART-Alpha encoders fail at this. A similar observation is in the third example where the model correctly captures the *"sharp suit"* and *"moustache"* as very nuanced details, unlike the rest of the models. These examples are direct evidence that our approach can truly capture the meaning of words in longer captions. These observations suggest that our TULIP text encoder enhances long caption comprehension, resulting in more accurate and contextually rich image generation across varying prompt lengths. We provide additional image generation examples and human evaluation results in the appendix.

## 4.3 ABLATION STUDY

In this section, we carefully analyze various design choices for our proposed method. We examine different positional encoding schemes, investigate the impact of context length, and explore variations in the distillation loss function.

**Different types of positional encodings.** In this experiment, we compare different types of positional encodings, namely absolute and relative ones. We choose a recently introduced Contextual Position Encoding (CoPE) (Golovneva et al., 2024) for implementing the relative positional encodings in the text encoder. Afterwards, we perform the distillation and context length extension phases using the same dataset and parameters as with RoPE (Su et al., 2024). In Table 3 we observe that RoPE outperforms CoPE in long-caption retrieval tasks. This is due to RoPE's strong ability to generalize across varying or extended sentence lengths, even beyond those on which the model was originally trained. In contrast, CoPE struggles to generalize as effectively when sequence lengths increase, as its embeddings are more dependent on the specific context within which they were trained. This explains why CoPE performs similarly to RoPE on the ShareGPT4V test split (same training distribution), but shows a larger performance gap on the Long-DCI and Urban-1k datasets.

**The impact of the caption length.** Next, we evaluate the impact of varying context lengths on the performance of RoPE in long-caption image retrieval tasks. To investigate this, we fine-tune only the text encoder during the context length extension phase with different context sizes (which are $n \times 77$): $\{77, 154, 231, 308\}$ tokens while keeping the image encoder frozen. We deliberately freeze the image encoder to isolate the text encoder's performance, as unfreezing it could potentially mask the text encoder's limitations in processing longer inputs. Figure 4 presents the model's performance across three datasets (ShareGPT4V, Urban-1K, and Long-DCI) for cross-modal retrieval tasks. We observe general improvement in performance with increased context length, particularly from 77 to 154 tokens, across all datasets. This improvement is more pronounced for image-to-text retrieval, suggesting that longer contexts enhance text representation and enable more precise alignment with image features. However, we observe a slight performance plateau or minor decline for 308-length sentences, signaling a point of diminishing returns where additional tokens may introduce noise or redundancy. This plateau is likely due to the average caption length in our training data being 174.02 tokens, explaining why performance levels off between 154 and 231 tokens.

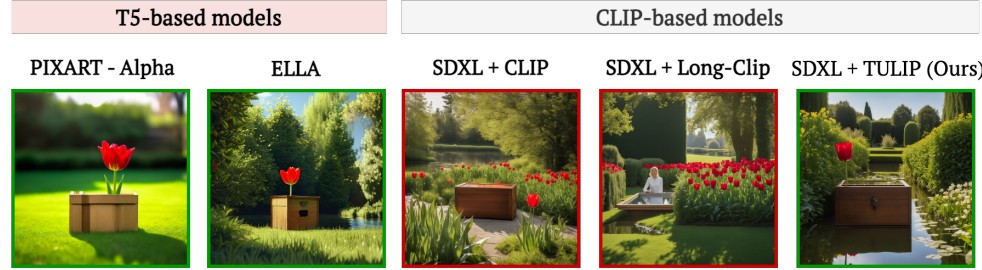

| T5-based models | | CLIP-based models | | |
|---|---|---|---|---|
| PIXART - Alpha | ELLA | SDXL + CLIP | SDXL + Long-Clip | SDXL + TULIP (Ours) |

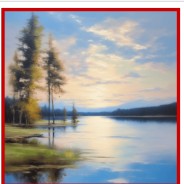 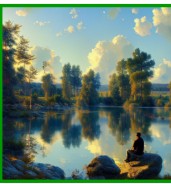 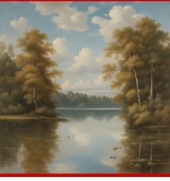 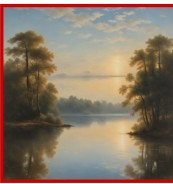 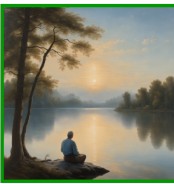

In a quiet garden, with tall green bushes on the left and a clear pond on the right, a wooden box sits on the grass in the middle. The box is slightly open, and inside is a bright red tulip, standing tall. The sunlight shines on the tulip, making it stand out against the smooth wood of the box. A light breeze moves through the garden, // but the tulip stays still, standing tall inside the box.

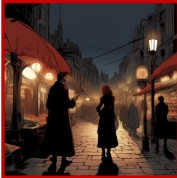 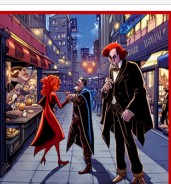 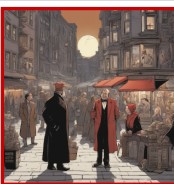 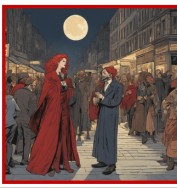 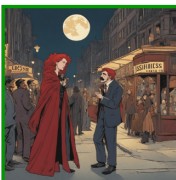

The painting captures a serene moment in nature. At the center, a calm lake reflects the sky, its surface rippled only by the gentlest of breezes. The sky above is a brilliant mix of blues and whites, with fluffy clouds drifting leisurely across. On the banks of the lake, tall trees stand gracefully, their leaves rustling in the wind. // The soft light of the setting sun bathes the entire scene in a warm glow, creating a sense of peace and tranquility. The colors are muted yet vibrant, and the details are captured with precision, giving the painting a sense of realism while still retaining a dreamlike quality. In the foreground, an old man sits on a rock, seemingly lost in deep thought or meditation.

A illustration from a graphic novel. A bustling city street under the shine of a full moon. The sidewalks bustling with pedestrians enjoying the nightlife. At the corner stall, a young woman with fiery red hair, dressed in a signature velvet cloak, is haggling with the grumpy old vendor. the grumpy vendor, a tall, sophisticated man is wearing a sharp suit, sports a // noteworthy moustache is animatedly conversing on his steampunk telephone.

Figure 3: **Text-to-Image Generation results.** We replace the text encoder of SDXL with our own TULIP model. We observe improvements in capturing nuanced details, compared to T5-based models such as PIXART-Alpha and ELLA, as well as CLIP-ViT-L-14 and Long-CLIP. Note that the // marks the 77-token boundary in the caption. The words in green indicate visual concepts that are correctly generated by SDXL + TULIP and are missed by the baselines.

| | Long-DCI | | ShareGPT4V | | Urban-1K | |
|---|---|---|---|---|---|---|
| Positional encodings | Img2Txt | Txt2Img | Img2Txt | Txt2Img | Img2Txt | Txt2Img |
| Absolute | 41.9 | 40.0 | 96 | 93.8 | 72.9 | 69.4 |
| CoPE | 50.8 | 49.9 | 98.5 | 97.8 | 86.7 | 82.8 |
| **RoPE** | **55.7** | **56.4** | **99.0** | **99.0** | **90.1** | **91.1** |

Table 3: **Ablation comparing different positional encodings such as Absolute, RoPE, and CoPE in TULIP.** RoPE generalizes better across varying or extended sentence lengths, especially on out-of-distribution datasets, namely Long-DCI and Urban-1k.

**Benefit of using cosine distillation loss.** In this section, we vary the distillation loss function used during the relative position adaptation phase and report the results in Table 4. As seen, the cosine loss performs better than other alternatives across different datasets and tasks. We believe this is attributed to its alignment with the normalized embedding space of CLIP models and its scale invariance property. The scale invariance of cosine loss proves particularly beneficial in this distillation setup, where the student model (ViT-L/14 with the added RoPE or CoPE layer) can produce embeddings with different magnitudes compared to the teacher model. This invariance

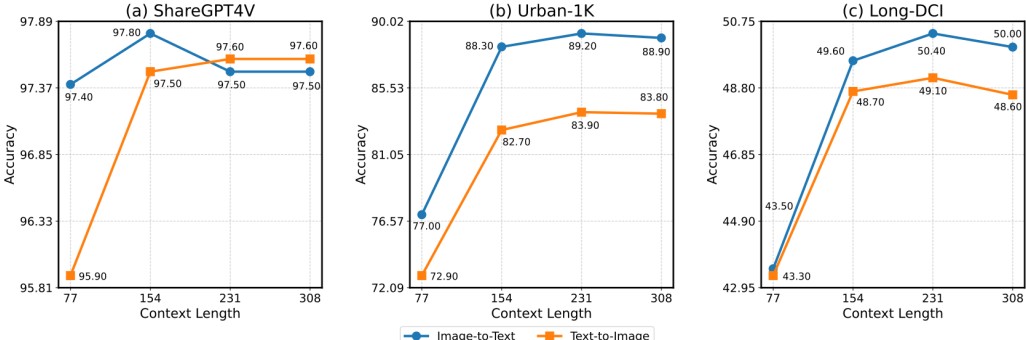

Figure 4: **Impact of the sequence length on cross-modal retrieval tasks**. We observe general improvement in performance with increased sequence length, particularly from 77 to 154 tokens, across all datasets and tasks.

allows the distillation process to focus on transferring the essential directional information of the embeddings, rather than being influenced by potential scale discrepancies introduced by the RoPE or CoPE layer. As a result, cosine loss consistently outperforms other loss functions like L2 and MSE across various datasets (Long-DCI, ShareGPT4V, Urban-1K) and cross-modal tasks such as image-to-text and text-to-image, demonstrating its effectiveness in preserving the semantic relationships learned by the teacher model.

| Distillation loss | Long-DCI | | ShareGPT4V | | Urban-1K | |
|---|---|---|---|---|---|---|
| | Img2Txt | Txt2Img | Img2Txt | Txt2Img | Img2Txt | Txt2Img |
| CLIP | 35.0 | 37.0 | 81.8 | 84.0 | 68.7 | 52.8 |
| L2 | **39.4** | 35.8 | 84.8 | 83.9 | 70.2 | 56.3 |
| MSE | 36.4 | 31.8 | 83.0 | 81.3 | 74.0 | 53.9 |
| **Cosine** | 38.5 | **35.8** | **84.8** | **84.2** | **73.6** | **56.6** |

Table 4: **Ablation comparing different distillation loss terms in TULIP.** Cosine loss yields the best performance across different datasets and tasks. Note that here we report the performance of the distilled models before the relative position expansion phase.

## 4.4 ADDITIONAL ANALYSIS

**Attention spread visualization.** The effectiveness of our model in long cross-modal retrieval tasks and image generation is largely due to its distinct attention distribution patterns. To illustrate this, we visualize the attention scores between the CLS text token and its preceding tokens in the final attention block of the text encoder, as shown in Figure 5. For comparison, we provided attention visualizations for both our TULIP and LongCLIP. This analysis reveals two key advantages of our model when processing a 248-token caption. First, our model exhibits a more uniform distribution of attention across input tokens, demonstrating its ability to expand the attention scope and effectively aggregate information from a broader range of tokens. This expanded attention field improves the model's performance in long-caption tasks by capturing and utilizing details from later parts of the caption that other models might overlook. Second, our model shows increased attention to punctuation symbols, particularly commas, which enhances its ability to parse and segment longer texts. Such capability is crucial for better comprehension of complex, multi-caption descriptions.

**Caption-image relevance distribution analysis.** This analysis investigates where relevant information is distributed within long captions given an image. The aim is to highlight how useful content is spread throughout the whole caption. We analyze 100 randomly selected images with long captions from the ShareGPT4V dataset. For each image-caption pair, we computed the visual embeddings of the image and compared them to the text embeddings of the caption subwindow. We define sliding subwindows of sizes 20, 33, and 55 tokens, moving from left to right with strides of 5, 10, and 15 tokens respectively. We then calculated the cosine similarity between the image encoding and the text encodings of each subwindow, visualized in Figure 6. The figure shows that the similarity scores are distributed across different subwindows, emphasizing the need for models capable of processing longer input sequences. Notably, as window size increases, the similarity patterns become more

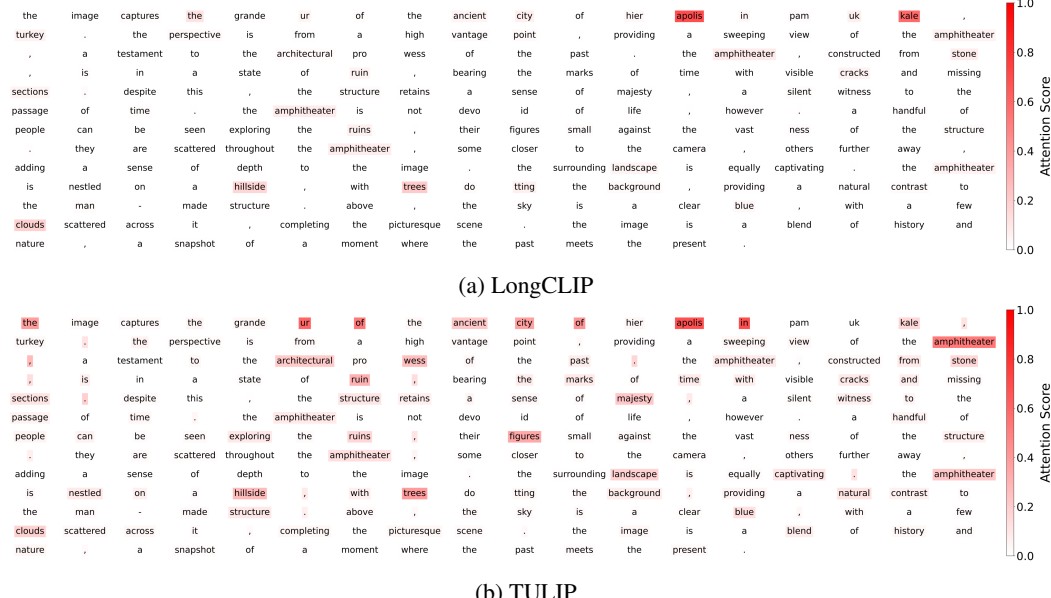

Figure 5: **Attention Spread Visualization** comparing (a) LongClip and (b) TULIP. Our model achieves uniform attention across tokens, demonstrating superior capabilities in parsing and segmenting longer texts with precision.

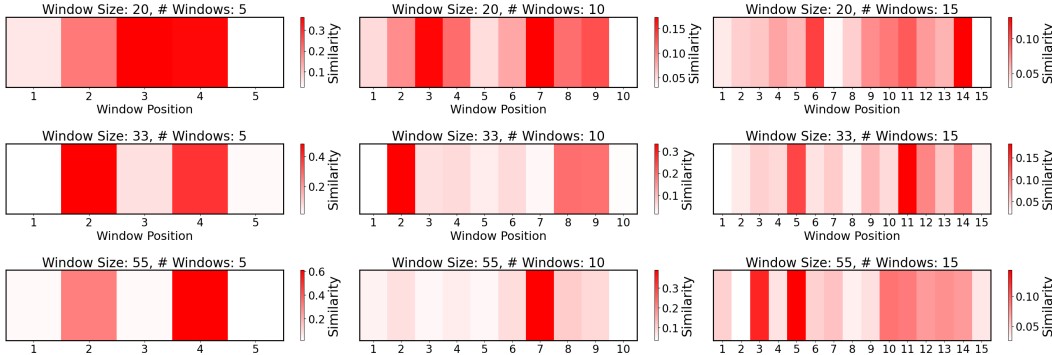

Figure 6: **Caption-image relevance distribution analysis** across varying window sizes and positions. It shows that image-relevant information is spread throughout captions, emphasizing the need for models to process longer text inputs to capture all pertinent details.

concentrated and pronounced, suggesting that larger context windows capture more cohesive and relevant information. Additionally, the variability in similarity across different windows highlights the non-uniform distribution of image-relevant information throughout the captions. This reinforces the need to leverage the entire textual sequence when learning image-text representations.

## 5 CONCLUSION

In this work, we addressed the limitations of CLIP-like models in handling long input sequences. We introduce TULIP, a generalizable method that upgrades the context length beyond the 77-token limit. By leveraging relative positional encodings, our approach enables the effective modeling of pairwise token relationships. Through a two-step training process, we successfully adapt the CLIP-like model to process longer captions without compromising its performance on shorter inputs. Our experiments demonstrate that TULIP considerably improves long-caption performance on cross-modal tasks such as retrieval and text-to-image generation, setting a new standard for vision-language contrastive models that require handling complex, extended text descriptions.

**Limitations.** Our reliance on ShareGPT4V dataset, synthesized by GPT4V (Achiam et al., 2023), limits TULIP's performance to the quality of GPT4V's long captions. Furthermore, while TULIP is theoretically capable of handling longer contexts due to the nature of the relative positional encodings, its token length is constrained by the average token length of the ShareGPT4V captions.

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

# A  APPENDIX

## A.1  QUALITATIVE EVALUATION OF IMAGE GENERATION USING T5-BASED MODELS, CLIP, LONG-CLIP, AND TULIP

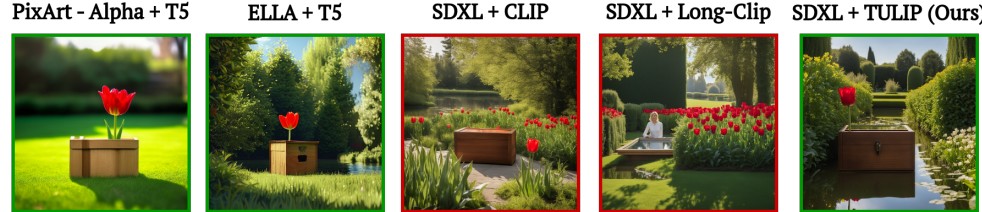

In a quiet garden, with tall green bushes on the left and a clear pond on the right, a wooden box sits on the grass in the middle. The box is slightly open, and inside is a bright red tulip, standing tall. The sunlight shines on the tulip, making it stand out against the smooth wood of the box. A light breeze moves through the garden, // but the tulip stays still, standing tall inside the box

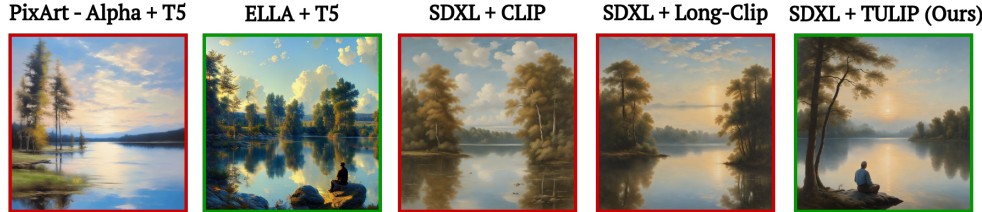

The painting captures a serene moment in nature. At the center, a calm lake reflects the sky, its surface rippled only by the gentlest of breezes. The sky above is a brilliant mix of blues and whites, with fluffy clouds drifting leisurely across. On the banks of the lake, tall trees stand gracefully, their leaves rustling in the wind. // The soft light of the setting sun bathes the entire scene in a warm glow, creating a sense of peace and tranquility. The colors are muted yet vibrant, and the details are captured with precision, giving the painting a sense of realism while still retaining a dreamlike quality. In the foreground, an old man sits on a rock, seemingly lost in deep thought or meditation.

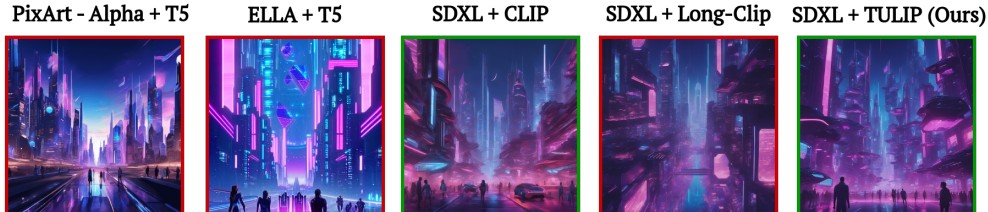

A futuristic city at night, illuminated by neon lights in various shades of blue, pink, and purple, with towering skyscrapers featuring unique geometric designs, flying cars zipping through the air, // and people walking along the streets below, dressed in sleek, futuristic clothing.

Figure 7: **Qualitative results of image generation using text encoders based on T5 (PIXART-Alpha and ELLA), CLIP, Long-CLIP, and TULIP.** Our TULIP-based model demonstrates the ability to generate images with nuanced details from captions, offering a plug-and-play solution without requiring costly retraining.

## A.2  HUMAN EVALUATION OF TEXT-TO-IMAGE GENERATION

We design a human evaluation study by manually crafting captions and generating images by using Stable Diffusion XL (SDXL) with CLIP and TULIP-based text encoders. The resulting images are randomized, and annotators are tasked with selecting the image that is best aligned with the prompt (See Figure 9). They were instructed to evaluate alignment based on objects, their attributes, and their relationships. A total of 15 annotators reviewed 20 samples. We report the results in Table 5. As can be seen, the win ratios are 89% for the SDXL + TULIP model and 11% for the SDXL + CLIP

| PixArt - Alpha + T5 | ELLA + T5 | SDXL + CLIP | SDXL + Long-Clip | SDXL + TULIP (Ours) |

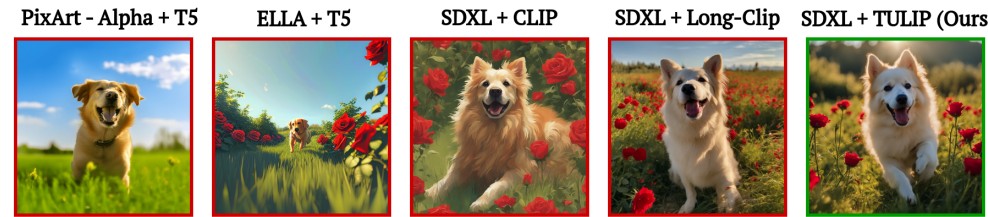

The sun was high in the clear blue sky, making the grass in the field look even greener as it swayed in the light breeze. A playful dog with soft, golden fur dashed across the grass, its paws kicking up tiny clumps of earth as it ran. Bright red roses stood tall in a nearby garden, their smooth petals glowing in the sunlight, adding a // burst of color to the scene.

| PixArt - Alpha + T5 | ELLA + T5 | SDXL + CLIP | SDXL + Long-Clip | SDXL + TULIP (Ours) |

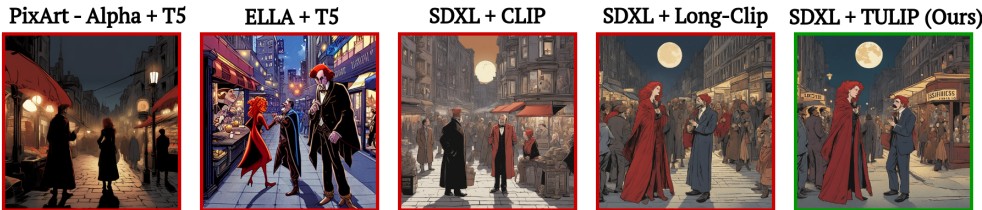

A illustration from a graphic novel. A bustling city street under the shine of a full moon. The sidewalks bustling with pedestrians enjoying the nightlife. At the corner stall, a young woman with fiery red hair, dressed in a signature velvet cloak, is haggling with the grumpy old vendor. the grumpy vendor, a tall, sophisticated man is wearing a sharp suit, sports a // noteworthy moustache is animatedly conversing on his steampunk telephone.

| PixArt - Alpha + T5 | ELLA + T5 | SDXL + CLIP | SDXL + Long-Clip | SDXL + TULIP (Ours) |

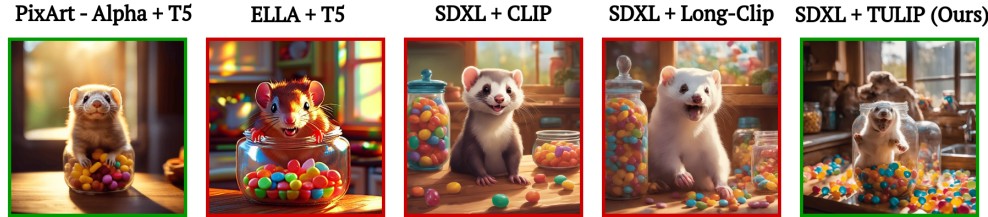

A mischievous ferret with a playful grin squeezes itself into a large glass jar, surrounded by colorful candy. The jar sits on a wooden table in a cozy kitchen, and warm sunlight filters through a nearby window. //

Figure 8: **Qualitative results of image generation using text encoders based on T5 (PIXART-Alpha and ELLA), CLIP, Long-CLIP, and TULIP.** Our TULIP-based model demonstrates the ability to generate images with nuanced details from captions, offering a plug-and-play solution without requiring costly retraining.

model, reflecting the enhanced alignment capability that we observe in the qualitative comparisons as well. The following instructions were provided to the annotators:

> **Instructions for annotators:**
> You are provided with 20 samples, each containing a prompt and two images generated by two different Stable Diffusion models. Your task is to compare the two images given the input prompt and choose the one you find more aligned with the prompt. Note that the differences might be very nuanced in some cases. Pay attention to the object attributes, relationships, verbs, object counting, etc.

| Method | Win rate (%) |
|---|---|
| SDXL + CLIP | 11 |
| **SDXL + TULIP (Ours)** | **89** |

Table 5: **Human evaluation results.** The results reflect the enhanced alignment capability that we observe in the qualitative comparisons as well.

Figure 9: **Examples used in the human evaluation study.** We provide the text prompt and two randomized images generated by SDXL with different variants of the text encoder (TULIP vs CLIP).

## A.3 EVALUATION OF COMPOSITIONAL UNDERSTANDING BENCHMARKS

CLIP-like models often struggle with accurately interpreting the compositional aspects of images, including spatial relationships, numeracy, and complex inter-object attributes. To evaluate TULIP's capabilities in these areas, we considered two well-established baselines for multimodal compositional understanding: ARO (Yuksekgonul et al., 2023) and VL-Checklist (Zhao et al., 2022). The results in Table 6 indicate that TULIP consistently outperforms CLIP and surpasses Long-CLIP in 4 out of 7 settings. These findings highlight TULIP's enhanced capability in handling fine-grained captions.

| Method | ARO | | | | VL-Checklist | | |
|---|---|---|---|---|---|---|---|
| | VGR | VGA | Flickr | COCO | Obj | Att | Rel |
| CLIP | 59.9 | 63.1 | 60.2 | 47.9 | 81.1 | 67.6 | 61.9 |
| Long-CLIP | **64.6** | **66.6** | 24.8 | 23.3 | 84.3 | 71.6 | **63.5** |
| **TULIP (Ours)** | 63.4 | 66.2 | **52.3** | **43.9** | **85.2** | **74.3** | 62.7 |

Table 6: **Comparison across compositional understanding benchmarks, namely ARO and VL-Checklist.** We can observe that TULIP consistently outperforms CLIP and surpasses Long-CLIP in 4 out of 7 settings, demonstrating its ability to handle fine-grained captions. Note that we use CLIP-ViT-L-14 as vision encoder.

## A.4 DOCCI BENCHMARK EVALUATION OF LONG-CAPTION RETRIEVAL

To additionally evaluate TULIP on long-caption retrieval tasks, we use DOCCI Onoe et al. (2024), which is a novel dataset consisting of images paired with long, human-annotated English descriptions. We performed experiments to compare our TULIP model to the baselines CLIP and Long-CLIP. We provide the top R@1 results in Table 7, where we observe that TULIP obtains the best performance, especially for image-to-text retrieval, demonstrating its efficiency and applicability across one more long-caption benchmark.

| | Img2Txt R@1 | Txt2Img R@1 |
|---|---|---|
| CLIP | 65.7 | 63.1 |
| Long-CLIP | 66.5 | 78.5 |
| **TULIP (Ours)** | **77.9** | **79.1** |

Table 7: **Long caption cross-modal retrieval comparison on DOCCI dataset.** We observe that TULIP obtains the best performance, especially for image-to-text retrieval, demonstrating its efficiency and applicability across one more long-caption benchmark.

## A.5    TRAINING DATA AMOUNT FOR TULIP STUDENT MODEL

To analyze the amount of data needed to train the student model, we conduct experiments using 33%, 66%, and the entire ShareGPT4V dataset (1.2 million image-caption pairs). The results on the retrieval tasks (top R@1) shown in Table 8, emphasize the importance of utilizing the full dataset, particularly for improving the performance on out-of-distribution data i.e. Long-DCI and Urban1k.

| | **Long-DCI** | | **ShareGPT4V** | | **Urban-1K** | |
|---|---|---|---|---|---|---|
| **Amount of data** | Img2Txt | Txt2Img | Img2Txt | Txt2Img | Img2Txt | Txt2Img |
| 33% | 15.2 | 12.2 | 88.6 | 90.0 | 33.6 | 36.9 |
| 66% | 28.7 | 26.2 | 94.6 | 95.5 | 37.6 | 45.0 |
| **100%** | **55.7** | **56.4** | **99.0** | **99.0** | **90.1** | **91.1** |

Table 8: Performance comparison across different dataset sizes.

## A.6    ADDITIONAL IMAGE GENERATION COMPARISON BETWEEN TULIP AND T5-BASED MODELS

The T5 text encoder (Roberts et al., 2019) is well-known for its ability to handle long captions. To evaluate how our TULIP method compares to T5-based approaches, we conducted additional text-to-image generation experiments. In particular, we provide a qualitative comparison between our TULIP-based model and two T5-based models, PIXART-Alpha (Chen et al., 2023a) and ELLA (Hu et al., 2024), in Figures 10, 11, 12, 13, 14 and 15. This comparison demonstrates that all models perform on par in generating high-quality images from long and nuanced text descriptions. However, our TULIP approach stands out due to its plug-and-play capability for image generation, requiring no additional fine-tuning, unlike PIXART-Alpha (Chen et al., 2023a) and ELLA (Hu et al., 2024). This characteristic makes TULIP more versatile and practical compared to the fine-tuning demands of the T5-based counterparts.

## A.7    QUALITATIVE EVALUATION OF CROSS-MODAL RETRIEVAL USING CLIP AND TULIP

Figures 16 and 17 showcase selected results from our qualitative evaluation of cross-modal retrieval tasks. We can observe that our TULIP model excels at capturing fine-grained details in both captions and images, outperforming the baseline CLIP.

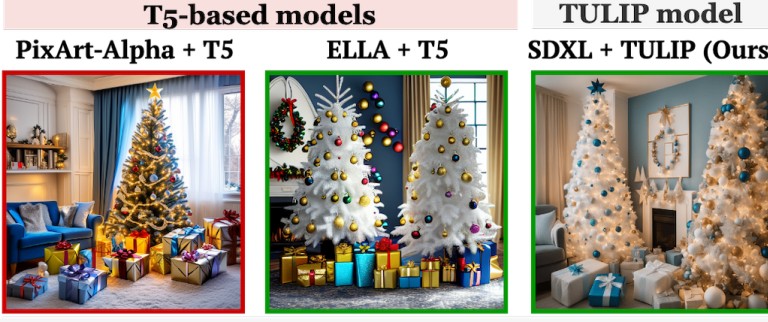

Two tall white Christmas trees, adorned with twinkling lights and colorful ornaments, stand in the corner of a cozy living room. Beneath the trees, wrapped presents in shiny blue and golden paper and colorful ribbons are stacked in a small mountain.

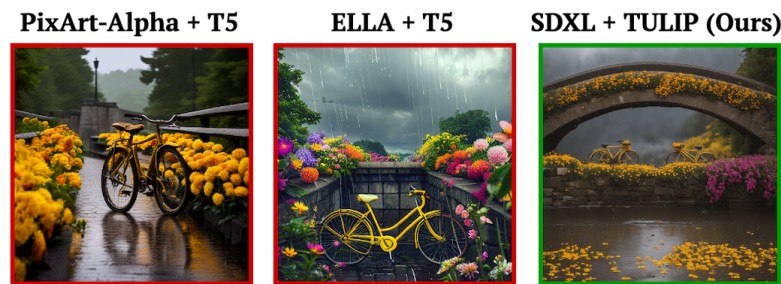

Colorful flowers in full bloom line the base of a stone bridge, their petals wet from the light drizzle that has begun to fall. A pair of yellow gold bikes rest on the stone edge, their bright, shiny surfaces reflecting the overcast light. As the thunderstorm looms closer, the wind picks up, causing the flowers to bend slightly. The storm's dark clouds create a dramatic backdrop, making the bikes and flowers stand out even more against the approaching gloom.

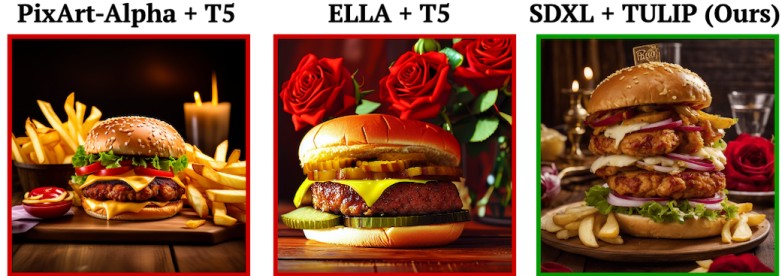

A juicy double chicken burger, piled high with pickles, onions, and a perfectly cooked patty, is placed on a wooden table, the bun soft and golden. A pile of fries, crispy on the outside and fluffy on the inside, sits beside the burger, their golden color shining in the light. A bunch of red roses, their petals open and lush, sits in the background, adding a touch of elegance to the scene. The warm, inviting colors of the food contrast beautifully with the deep red of the roses.

Figure 10: Comparison to T5-based models (PIXART-Alpha (Chen et al., 2023a) and ELLA (Hu et al., 2024)) for image generation.

| T5-based models | | TULIP model |
|---|---|---|
| **PixArt-Alpha + T5** | **ELLA + T5** | **SDXL + TULIP (Ours)** |

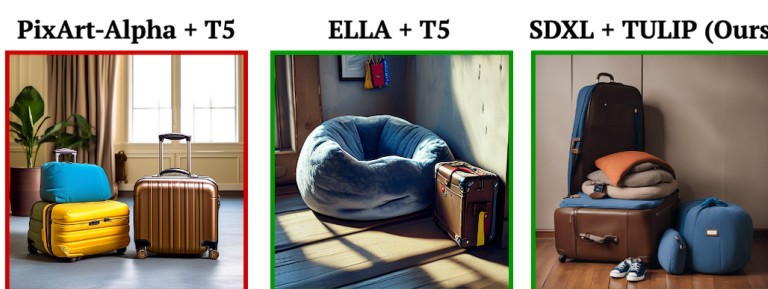

A tall martini glass with a sleek, clear stem holds a perfectly mixed drink, the glass gleaming under the soft light. Inside the glass, a single green olive floats at the base, its shiny surface reflecting the light. Nearby, a plate of fresh oysters is arranged neatly, their glossy, wet shells slightly open, revealing the tender meat inside. The contrast between the elegant martini and the rustic oysters creates a striking visual pairing on the table.

**PixArt-Alpha + T5**    **ELLA + T5**    **SDXL + TULIP (Ours)**

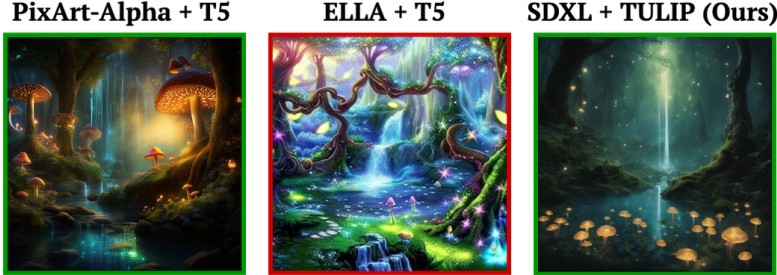

A sturdy brown suitcase with brass clasps rests on the floor next to a soft, round bean bag. The suitcase is slightly scuffed, showing signs of travel, with a colorful luggage tag hanging from the handle. The bean bag, filled with plush, squishy foam, is covered in a cozy blue fabric, inviting someone to relax. The contrast between the hard, structured suitcase and the soft, relaxed shape of the bean bag creates a welcoming atmosphere.

**PixArt-Alpha + T5**    **ELLA + T5**    **SDXL + TULIP (Ours)**

An enchanted forest at dawn, where glowing mushrooms light up the forest floor, trees with twisted trunks and sparkling leaves reach towards the sky, and magical creatures like fairies and wisps float through the air, while a waterfall cascades into a crystal-clear pond in the background.

Figure 11: Comparison to T5-based models (PIXART-Alpha (Chen et al., 2023a) and ELLA (Hu et al., 2024)) for image generation.

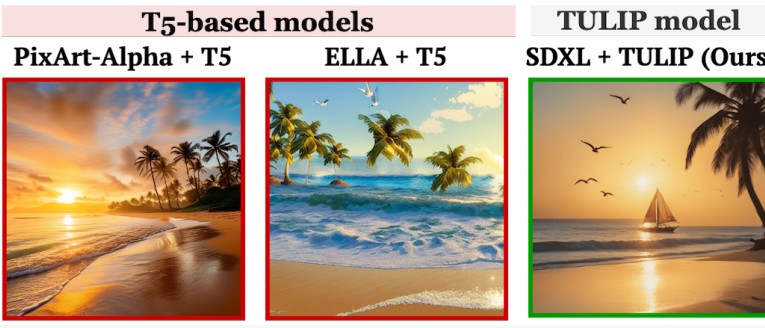

A serene beach at dawn, where the waves gently lap against the shore, the sand soft and golden underfoot, and palm trees sway in the light breeze, while seagulls call overhead. A lone sailboat drifts on the horizon as the sun rises, casting a golden light across the water.

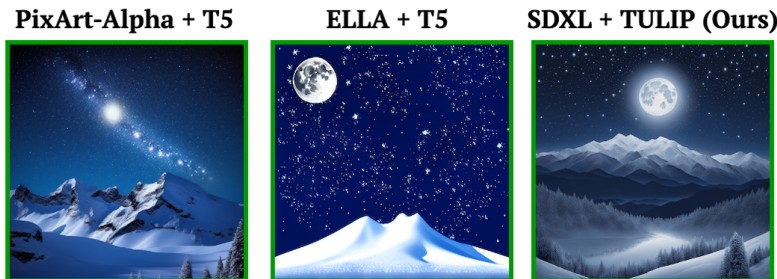

A bright, full moon hangs high in the sky, its glowing surface casting a soft, silvery light over the landscape. The starry night is dotted with countless sparkling stars, their brilliance forming delicate constellations against the deep navy backdrop. Below, a single snowy mountain rises majestically, its peak glistening under the moonlight. The crisp white snow contrasts sharply with the dark sky, creating a serene and magical scene

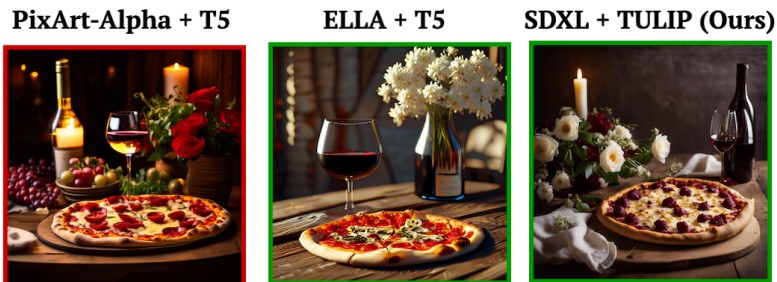

A freshly baked pizza with cheese and vibrant toppings sits on a wooden table, its crust golden and crisp. Beside it, a glass of deep red wine rests, its smooth surface reflecting the soft glow of candlelight. A small vase of white flowers sits nearby, their delicate petals bright and pure against the rustic table setting. The contrasting textures of the warm pizza, the wine glass, and the soft blooms create a harmonious and inviting scene.

Figure 12: Comparison to T5-based models (PIXART-Alpha (Chen et al., 2023a) and ELLA (Hu et al., 2024)) for image generation.

| **T5-based models** | | **TULIP model** |
| --- | --- | --- |
| **PixArt-Alpha + T5** | **ELLA + T5** | **SDXL + TULIP (Ours)** |

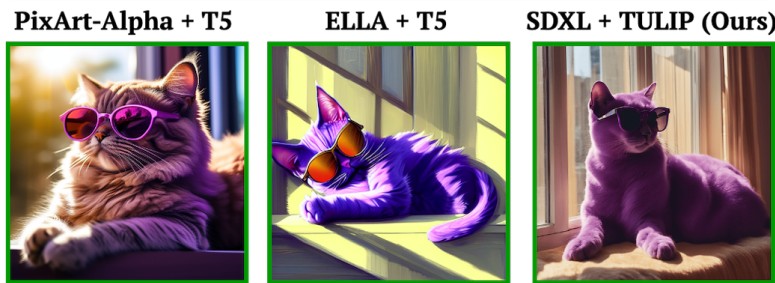

A glass vase filled with fresh pink tulips sits in the center of a desk, the long green stems visible through the crystal-clear surface. The petals of the tulips are soft, their delicate pink hue adding warmth to the room. Next to the vase, a pair of neatly stacked books lies with one slightly ajar, revealing a glimpse of printed pages. The arrangement of flowers and books creates a serene, cozy corner ideal for reading or reflection.

| **PixArt-Alpha + T5** | **ELLA + T5** | **SDXL + TULIP (Ours)** |
| --- | --- | --- |

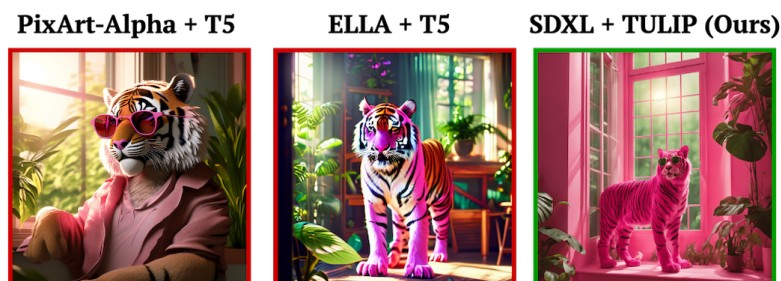

The purple cat lounges lazily on a sunny windowsill, its bright fur standing out against the light-colored wood. Its oversized sunglasses reflect the sunlight, adding a cool, fashionable touch to its relaxed pose.

| **PixArt-Alpha + T5** | **ELLA + T5** | **SDXL + TULIP (Ours)** |
| --- | --- | --- |

The pink tiger stands gracefully in the corner of the house with a green sun glasses, and its vibrant fur shimmering in the sunlight streaming through the window. Its intense gaze blends with the calm atmosphere of the room, surrounded by plants and cozy furniture.

Figure 13: Comparison to T5-based models (PIXART-Alpha (Chen et al., 2023a) and ELLA (Hu et al., 2024)) for image generation.

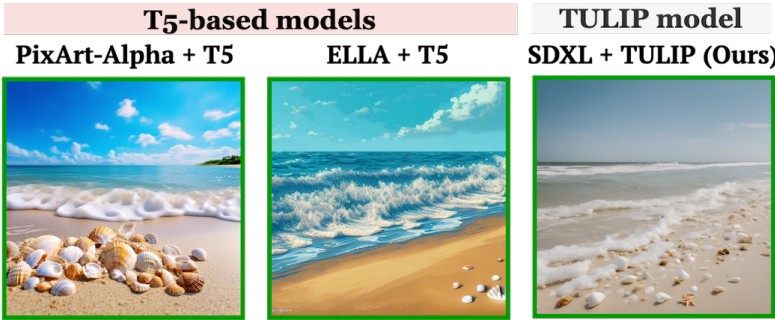

A golden sandy beach stretches out under a blue sky, with gentle waves lapping at the shore. A few white seashells are scattered along the edge.

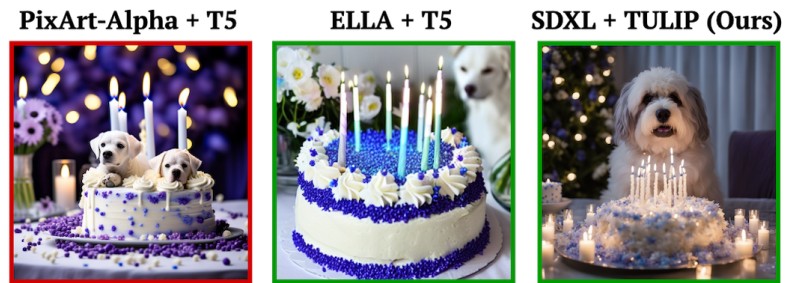

A birthday cake with creamy white frosting is decorated with blue and purple sprinkles and topped with lit candles. The cake sits on a table surrounded by white flowers next to a white dog.

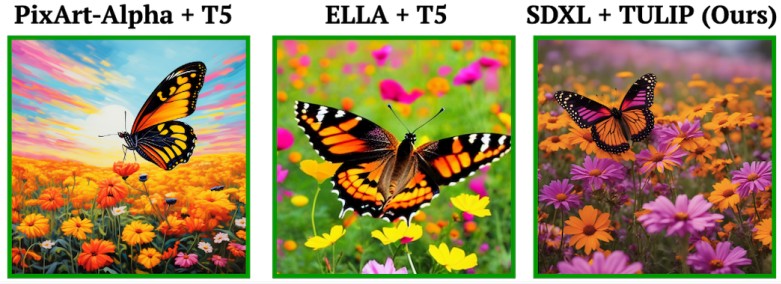

A butterfly with orange and black wings flies gently over a vibrant flower field. The field is filled with colorful blooms of yellow and pink, their petals swaying in the breeze.

Figure 14: Comparison to T5-based models (PIXART-Alpha (Chen et al., 2023a) and ELLA (Hu et al., 2024)) for image generation.

**T5-based models**       **TULIP model**

**PixArt-Alpha + T5**     **ELLA + T5**     **SDXL + TULIP (Ours)**

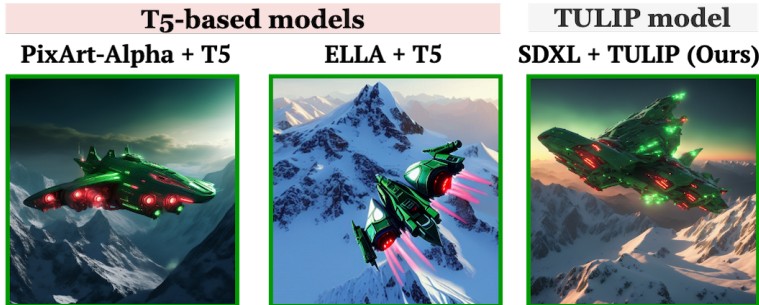

A green spaceship with glowing red engines flies above a snowy mountain. The mountain's jagged peaks glisten with fresh snow under the soft light of dawn.

**PixArt-Alpha + T5**     **ELLA + T5**     **SDXL + TULIP (Ours)**

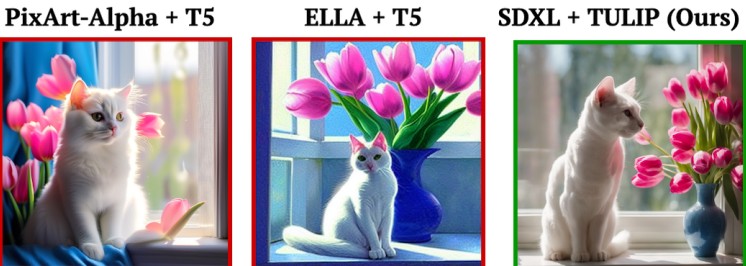

A white cat sits on a windowsill beside a vase filled with bright pink tulips. The cat's eyes are on the petals as sunlight filters through the blue vase.

Figure 15: Comparison to T5-based models (PIXART-Alpha (Chen et al., 2023a) and ELLA (Hu et al., 2024)) for image generation.

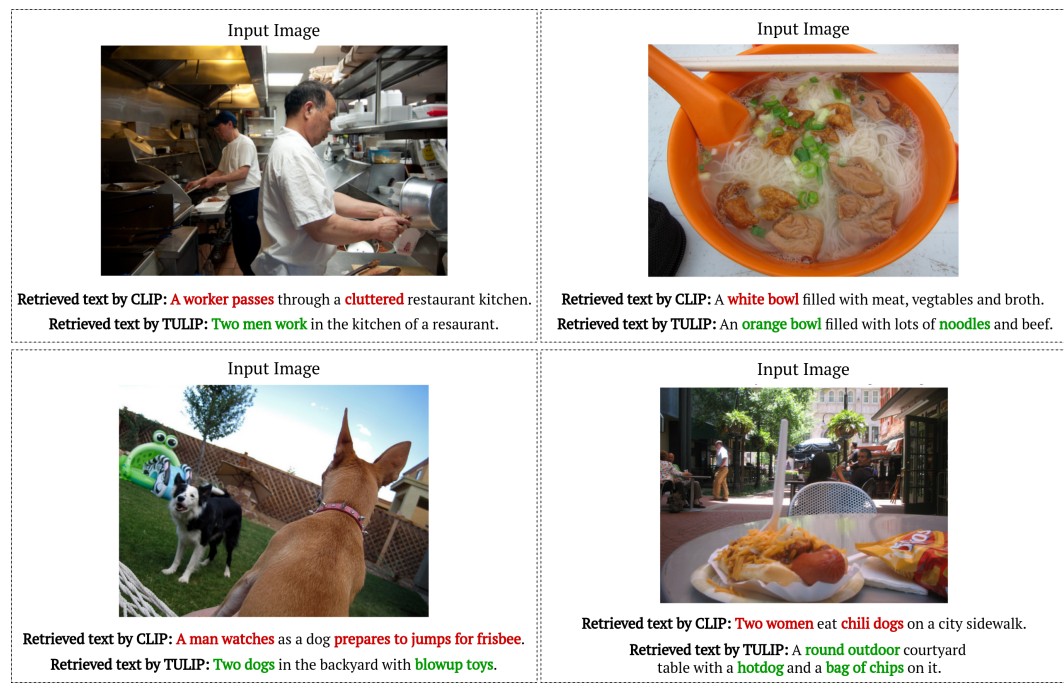

Figure 16: **Qualitative comparison of image-to-text retrieval between CLIP and TULIP.** We can observe that our TULIP model can capture the fine-grained details in both captions and images. Note that the red text color indicates visually misclassified concepts in the images, whereas the green color indicates correct ones.

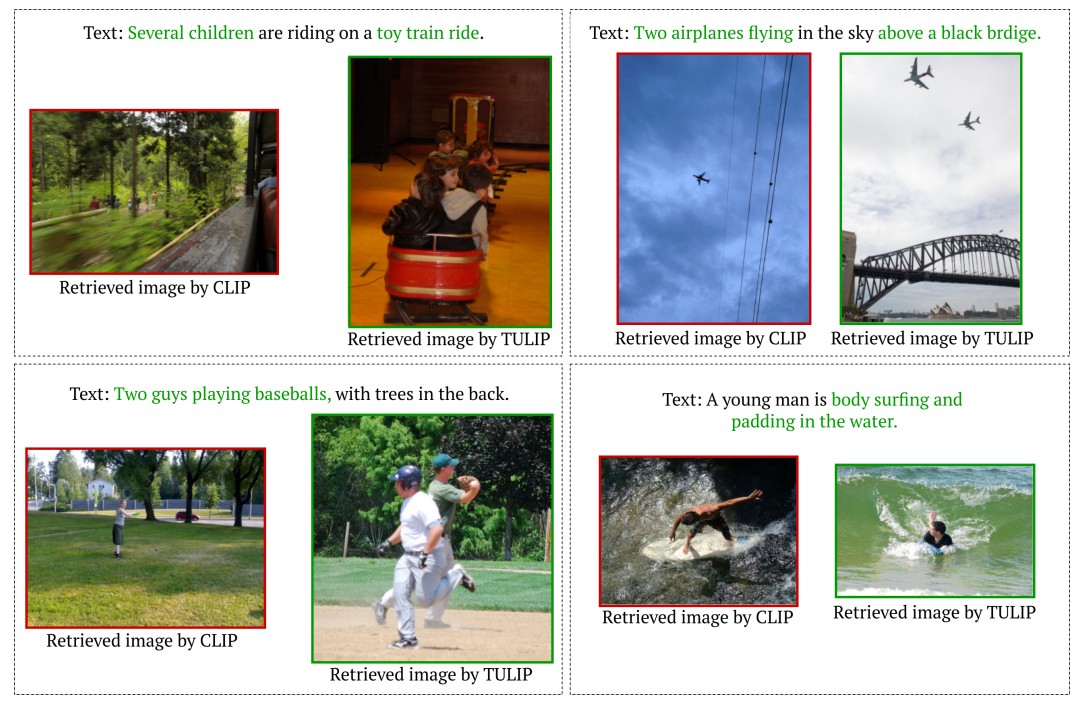

Figure 17: **Qualitative comparison of text-to-image retrieval between CLIP and TULIP.** We can observe that our TULIP model can capture the fine-grained details in both captions and images. Note that the red text color indicates visually misclassified concepts in the images, whereas the green color indicates correct ones.

