# OpenReview forum: "TULIP: Token-length Upgraded CLIP"
_ICLR.cc/2025/Conference — ICLR 2025 Poster_

### Official Review · Reviewer_FB4m · 2024-11-03

**Soundness:** 3
**Presentation:** 3
**Contribution:** 2
**Rating:** 8
**Confidence:** 3

**Summary:**

The authors propose TULIP, a method that allows for extending the capabilities of CLIP beyond the 77 token limit that is often used when training such models. The authors demonstrate that their method improves performance of CLIP trained models, on both traditional image-text retrieval tasks, as well as long-context tasks (one of which is proposed by the authors themselves).

**Strengths:**

- The paper examines the quality of CLIP models in the setting of long text descriptions of images. This setting is of interest to the community - as the context of models grows, the ability to retain information from the entire caption decreases as well, and trying to remedy this is an interesting problem.

- The paper is also clear and easy to understand. I did not have any issues understanding the method presented, or any other part of the paper.

- I also believe that the inclusion of the Long-DCI evaluation dataset is an important contribution to the community. Having benchmark specifically for images with long context captions is extremely important in the advancement of this field.

**Weaknesses:**

- While the method itself appears useful, I am slightly worried about its novelty. At the end of the day, the model still requires training with longer contexts, so it is not immediately clear to me how different the method is from prior work. I believe that elaborating a bit more on which part of the method proposed by the authors provides the most benefit over prior work such as Long-CLIP would alleviate this concern.

- The paper has an important limitation, in that the setting considered is only CLIP-style models, and not autoregressively trained ones. The latters tend to have much larger context length than the ones considered in this paper. While I understand that direct comparison between the two may not be possible/ideal, having the method being applied to this category of models would greatly strengthen the paper.

- I would also be grateful if the authors could perform evaluation on a task based on the DOCCI dataset [A], which has the same stated goal as the proposed Long-DCI one.

- The experimental evaluation on image generation in Section 4.2 is qualitative rather than quantitative. I believe that the authors could greatly strengthen the conclusions of the paper by evaluating the generated images in a more principled manner (for example, by performing a human preference study on the generated images).

Reference:

[A] Onoe et al., "DOCCI: Descriptions of Connected and Contrasting Images", ECCV 2024

**Questions:**

I would be grateful if, in addition to the points I raised above, the authors could elaborate on the amount of data with long context is needed to train the student model with TULIP, in order for it to perform well on long contexts.

---

> ### Author Response · Authors · 2024-11-22
> **Response to Reviewer FB4m**
>
> We thank Reviewer FB4m for the constructive feedback and insightful comments. With the responses below, we hope to address the Reviewer's concerns.
>
> **Contributions, w.r.t prior work.** A key difference of our approach w.r.t prior work, such as Long-CLIP (Zhang et al., ECCV 2024), is the use of relative positional encodings. Long-CLIP is performing stretching of the fixed absolute encodings, which is an ad-hoc solution since it is not scalable to longer contexts and is bounded to 248 tokens. Unlike them, our proposed solution is generalizable since we lift the constraint on fixed encodings and leverage the pairwise distance between tokens despite the length. In this manner, our approach can accommodate variable token lengths, ranging from the original 77 tokens to 300 tokens and beyond, as shown in Figure 4.
>
> **Comparison to autoregressive models.** Our work aims to explore why CLIP, a basis for most foundation models, lags behind autoregressive models in long-context understanding. One reason why autoregressive models excel in this area is due to their use of relative positional encodings. We adapt these principles to CLIP and will clarify this focus further in the Introduction section.
>
> **Evaluation on DOCCI.** We have performed the suggested experiments to evaluate our TULIP model on DOCCI, as well as the prior works CLIP and Long-CLIP. We provide the top R@1 results below and will add them to the paper. We observe that TULIP obtains the best performance, especially for img2txt retrieval, demonstrating its efficiency and applicability across one more benchmark.
>
> |           | Img2Txt R@1 | Txt2Img R@1 |
> |-----------|:-------------:|:-------------:|
> | CLIP      |   65.7  |   63.1  |
> | Long-CLIP |   66.5   |   78.5  |
> | **TULIP (Ours)**     |   **77.9**  |   **79.1**  |
>
> **Quantitative evaluation of image generation.** We designed a pilot human evaluation by manually crafting captions and generating images by using Stable Diffusion XL (SDXL) with CLIP and TULIP-based text encoders. The resulting images were randomized, and annotators were tasked with selecting the image that was best aligned with the prompt. They were instructed to evaluate alignment based on objects, their attributes, and their relationships. A total of 15 annotators reviewed 20 samples. The win ratios are 89% for the SDXL + TULIP model and 11% for the SDXL + CLIP model, reflecting the enhanced alignment capability that we observed in the qualitative comparisons as well.  We will add a discussion on this in the Experiment section of the paper.
>
> | Method   | Win rate (%)  |
> | ------------ | ------------ |
> | SDXL + CLIP   | 11  |
> | **SDXL + TULIP (Ours)**  |  **89** |
>
> **Amount of data for TULIP student model.** To analyze the needed amount of data with long context, we conduct experiments to train the student model using 33%, 66%, and the entire ShareGPT4V dataset (1.2 million image-caption pairs). The results on the retrieval tasks (top R@1) shown in the table below, emphasize the importance of utilizing the full dataset, particularly for improving the performance on out-of-distribution data i.e. Long-DCI and Urban1k. These findings will be included in the Experiments section of the paper.
>
> | Amount of data | Long-DCI I2T | Long-DCI T2I | ShareGPT4V I2T | ShareGPT4V T2I | Urban-1K I2T | Urban-1K T2I |
> |--------------------|--------------|--------------|----------------|----------------|--------------|--------------|
> | 33%                | 15.2         | 12.2         | 88.6           | 90.0           | 33.6         | 36.9         |
> | 66%                | 28.7         | 26.2         | 94.6           | 95.5           | 37.6         | 45.0         |
> | 100%               | **55.7**     | **56.4**     | **99.0**       | **99.0**       | **90.1**     | **91.1**     |
>
> We hope this addresses your concerns. Please let us know if you have any further questions or feedback.

---

> > ### Author Response · Authors · 2024-11-25
> > **Follow-Up on Rebuttal Questions**
> >
> > Dear Reviewer FB4m,
> >
> > Thank you once again for taking the time and effort to review our submission. As the deadline for the discussion phase is approaching, we wanted to kindly check if you have any additional questions or concerns that we could address in our rebuttal.
> >
> > We appreciate your insights and look forward to your feedback.
> >
> > Best regards,
> >
> > The Authors

---

> > ### Comment · Reviewer_FB4m · 2024-11-25
> > **Re: Response to Reviewer FB4m**
> >
> > Thank you for addressing both mine and the rest of the reviewers' comments, as well as the additional experiments.
> >
> > I would like to ask a few questions on the new results:
> >
> > - Would it be possible to include the detailed instructions given to the annotators in the paper (if there was a specific prompt given)? Also, it seems that the win rate is calculated across annotation instances - in my opinion it would be better if it was calculated across samples (with a "win" being classified as the majority of the annotators agreeing). That would however require a change in the experiments (more samples and potentially fewer annotators per sample), so I understand if it is not possible at this time.
> >
> > - In the ablation for the amount of data, how many passes of the data has each model been trained on? Is it only 1 pass?

---

> ### Author Response · Authors · 2024-11-25
> **Response to Reviewer FB4m**
>
> We thank the Reviewer for the engagement and suggestions. Below we provide our answers:
>
> **[Q1]** For the human evaluation, the instructions given to the annotators are as follows:
>
> *“Instructions: You are provided with 20 samples, each containing a prompt and two images generated by two different Stable Diffusion models. Your task is to compare the two images given the input prompt and choose the one you find more aligned with the prompt. Note that the differences might be very nuanced in some cases. Pay attention to the object attributes, relationships, verbs, object counting, etc.”*
>
> We added this in a separate section to the Appendix, as well as two samples included in the human evaluation experiment. Regarding the win rates, indeed they are calculated across annotations instances. We agree that calculating win rates across samples (defined by majority agreement) would provide a valuable perspective on the performance of our model. While it may not be feasible due to limited time, we plan to conduct an additional human study incorporating your suggestions to improve the robustness and interpretability of our results.
>
> **[Q2]** In the ablation for the amount of data, we indeed trained the models on 1 pass of the dataset to ensure a fair comparison.
>
> We hope this addresses your questions and welcome further comments or suggestions.

---

> ### Author Response · Authors · 2024-11-30
> **Follow-Up on Rebuttal Questions**
>
> Dear Reviewer FB4m,
>
> Thank you once again for your encouragement and thoughtful feedback. As the discussion deadline approaches, we wanted to follow up on our recent response to ensure we have addressed all your concerns. If there is anything further that might convince you of the value of our work, we would be happy to provide additional clarifications.
>
> Sincerely,
>
> The Authors

---

> > ### Comment · Reviewer_FB4m · 2024-12-01
> > **Re: Follow-Up on Rebuttal Questions**
> >
> > Thank you for the response to my follow-up questions. As most of my concerns have been alleviated, I am raising my score.

---

> > > ### Author Response · Authors · 2024-12-02
> > > **Thank you for the feedback and increased score**
> > >
> > > Dear Reviewer FB4m,
> > >
> > > Thank you once again for your feedback. We highly appreciate your continuous encouragement and the raised score.
> > >
> > > Sincerely,
> > >
> > > The Authors

---

### Official Review · Reviewer_3TVK · 2024-11-04

**Soundness:** 3
**Presentation:** 3
**Contribution:** 3
**Rating:** 6
**Confidence:** 4

**Summary:**

The TULIP model enhances CLIP's capability to process long captions by replacing fixed positional embeddings with relative positional encodings, extending the model's inherent context window beyond the 77 token limit. This approach is shown to improve performance on cross modal retrieval and text to image generation tasks. The TULIP training method is 2 stage - relative position distillation from the original CLIP model and finetuning to handle longer captions. Benchmark tests demonstrate TULIP's superior performance in long-caption scenarios. The paper also introduces a new evaluation benchmark, Long-DCI, to better assess long-caption retrieval tasks.

**Strengths:**

- Significant improvements across cross-modal retrieval and text-to-image generation
- The introduction of a benchmark for long captions is a step towards better evaluation of other research works in this domain
- The innovative approach with relative captioning and distillation has shown improved performance
- The paper writing is easy to follow

**Weaknesses:**

- Switching from absolute to relative positional encodings does improve flexibility with token length but can also introduce challenges in retaining fine-grained positional relationships in shorter contexts
- No specific details about the human annotators
- Needed a more qualitative comparison of how the new approach led to better performance in shorter context
- While CLIP possesses excellent zero-shot capabilities, it suffers from certain limitations in perceptual understanding. A discussion on TULIPs limitations in perceptual understanding would boost the presentation of the proposed approach

**Questions:**

1) How does the new model affect the compositionality of text inputs as dicussed here? https://aclanthology.org/2023.emnlp-main.301.pdf
2) What do you attribute the improved performance in a shorter context to? In Table 2, TULIP performs a lot better than CLIP.
3) Why are the values in Table 4 lower than those in Tables 1 or 3? If I missed something, can you clarify the caption to avoid confusion?

---

> ### Author Response · Authors · 2024-11-22
> **Response to Reviewer 3TVK**
>
> We thank Reviewer 3TVK for the constructive feedback and acknowledgment. We hope to address all comments below and will add the new experiments to the Experiments section of the paper.
>
> **Retaining fine-grained relationships in shorter contexts.**  Based on prior works (e.g. Llama 3, Gemma) and our empirical results, we believe that there is no evidence that switching from absolute to relative positional encodings poses a challenge in dealing with shorter contexts. Table 2 of the paper shows that TULIP not only retained the original performance of CLIP but even improved it by 6.1% R@1 on image-to-text retrieval and 10.7% R@1 on text-to-image retrieval on COCO. Similarly, we observed 8.2% R@1 on image-to-text retrieval and 13.6% R@1 on text-to-image retrieval on the Flickr dataset.
>
> **Details about human annotators.** The Long-DCI benchmark is derived from the dataset introduced in the paper “A Picture is Worth More Than 77 Text Tokens: Evaluating CLIP-Style Models on Dense Captions”, by Urbanek et al. (CVPR 2024). All human annotations were provided in this dataset, by using Mephisto (Urbanek and Ringshia, 2023) to host the task and pay crowdworkers to annotate the images, as detailed in Section 3.2 of their paper. Unlike their work, which uses only captions truncated to 77 tokens, we leverage their full annotations to construct a long-caption retrieval benchmark.
>
> **Qualitative comparison for short context.** To compare our TULIP model to CLIP, we perform a new qualitative analysis for short caption retrieval and present the results in the Appendix (Figures 12 and 13). We observe that our TULIP can also capture fine-grained details in short captions, such as object numeracy, relationships, attributes, etc. This is attributed to using relative positional encodings, which model pairwise token relationships unlike absolute ones, improving its understanding of complex details even in shorter contexts. Note that in the figure, the misclassified concepts are marked in red and the correct ones are in green.
>
> **Discussion on perceptual understanding and compositionality.** Indeed CLIP has challenges in perceptual understanding such as spatial understanding, numeracy, and compositionality of objects and attributes. To better evaluate TULIP's capabilities in these areas—specifically in handling compositionality in text inputs—we considered the two benchmarks suggested by the Reviewer (CompPrompt and ControlledImCaps). However, these were unfortunately not publicly available. Nevertheless, we turned to two well-established baselines for multimodal perceptual understanding and compositionality: ARO (Yuksekgonul et al. (ICLR 2023)) and VL-Checklist (Zhao et al. (EMNLP 2022)). The results, shown below, indicate that TULIP consistently outperforms CLIP and surpasses Long-CLIP in 4 out of 7 settings.
>
> | Model        | ARO VGR  | ARO VGA  | ARO FLICKR | ARO COCO | VLC Obj  | VLC Att  | VLC Rel  |
> |--------------|----------|----------|------------|----------|----------|----------|----------|
> | CLIP         | 59.9     | 63.1     | 60.2       | 47.9     | 81.1     | 67.6     | 61.9     |
> | Long-CLIP    | **64.6** | **66.6** | 24.8       | 23.3     | 84.3     | 71.6     | **63.5** |
> | **TULIP (Ours)** | 63.4     | 66.2     | **52.3**   | **43.9** | **85.2** | **74.3** | 62.7     |
>
> **Improved performance in shorter contexts (Table 2).** We attribute the improved performance on short-caption retrieval to using *relative positional encodings*. This allows TULIP to dynamically encode token relationships by modeling their dependencies without being constrained by fixed positions. This benefits both short and long contexts. To validate this, we perform an additional experiment to train TULIP with absolute encodings and compare it to its original version where we use relative ones. We present the top R@1 results below, where it can be observed that relative encodings indeed yield much better performance in short-caption retrieval for all four settings.
>
> | Method | COCO I2T | COCO T2I | Flickr I2T | Flickr T2I |
> |----------------------|----------|----------|------------|------------|
> | TULIP w/ Absolute  | 58.1     | 41.4     | 48.7       | 34.0       |
> | **TULIP w/ Relative (RoPE)**      | **62.6** | **46.1** | **56.7**   | **41.6**   |
>
> **Results in Table 4 compared to Tables 1 and 3.** Table 4 reflects the performance of the distilled models before the relative position expansion phase, whereas Tables 1 and 3 present results after full training is completed. This distinction was intentional, as Table 4 specifically aims to isolate and analyze the relative position distillation stage, allowing us to evaluate the effectiveness of different distillation losses. To address the potential confusion, we will revise the caption of Table 4 to clarify this point.
>
> We hope this addresses your concerns. We appreciate your consideration and hope this motivates you to upgrade your score. Please let us know if you have any further questions.

---

> > ### Author Response · Authors · 2024-11-25
> > **Follow-Up on Rebuttal Questions**
> >
> > Dear Reviewer 3TVK,
> >
> > Thank you once again for taking the time and effort to review our submission. As the deadline for the discussion phase is approaching, we wanted to kindly check if you have any additional questions or concerns that we could address in our rebuttal.
> >
> > We appreciate your insights and look forward to your feedback.
> >
> > Best regards,
> >
> > The Authors

---

> > > ### Comment · Reviewer_3TVK · 2024-11-27
> > > **Response to Follow-Up on Rebuttal Questions**
> > >
> > > Thank you for taking the time to reply to each question and running new experiments. I am satisfied with the responses and have revised the score from 5 to 6.

---

> > > > ### Author Response · Authors · 2024-11-27
> > > > **Thank you for the feedback and increased score**
> > > >
> > > > Dear Reviewer 3TVK,
> > > >
> > > > Thank you for reviewing our rebuttal and considering our additional experiments. We appreciate your thoughtful feedback and the increased score.
> > > >
> > > > Best regards,
> > > >
> > > > The Authors

---

### Official Review · Reviewer_XgPN · 2024-11-04

**Soundness:** 3
**Presentation:** 3
**Contribution:** 2
**Rating:** 6
**Confidence:** 4

**Summary:**

The paper addresses the challenge of integrating positional information effectively in contrastive vision-language models, particularly when dealing with long captions. Traditional models like CLIP are limited by short context windows, which restricts their ability to process detailed and dense textual descriptions. The authors propose an approach that combines Rotary Position Embedding (RoPE) and Contextual Position Encodings (CoPE) to better handle long captions without the need for training from scratch. This method aims to enhance the model's ability to comprehend pairwise token relationships and capture fine-grained relative positions in longer, more complex captions. The paper builds upon existing work such as Long-CLIP and proposes a refined technique to address the limitations of absolute positional encodings through interpolation.

**Strengths:**

1. Although the techniques themselves (RoPE and CoPE) are not novel, the paper demonstrates creativity in combining these methods to tackle the limitations of existing models like CLIP. This approach leverages the strengths of both RoPE and CoPE to address the shortcomings of absolute positional encodings, thereby providing a more dynamic and effective way to capture positional relationships in long sequences.

2. The paper is well-written and clearly presented. The authors effectively communicate their ideas and methodologies, making it accessible to readers with a background in machine learning and natural language processing. The use of figures, tables, and examples to illustrate key points is helpful in conveying the technical details of the proposed approach.

**Weaknesses:**

1. The proposed method appears to be an incremental improvement based on existing models like Long-CLIP. The combination of RoPE and CoPE, while potentially effective, does not introduce fundamentally new concepts. Both RoPE and CoPE are not novel and have been introduced in prior works. The primary contribution seems to be the application of these methods to the specific task of processing long captions, which may not be sufficient to claim significant novelty.

2. The paper does not adequately compare its proposed method with other approaches [1] that handle long text descriptions, such as those based on the encoder of T5 model [2]. T5-based models are known for their capability to process long texts effectively. The authors need to clearly articulate the advantages of their approach over T5-based methods and provide empirical evidence to support these claims. Without this comparison, it is challenging to assess the relative effectiveness of the proposed method.

3. The paper lacks a deep theoretical analysis of why the combination of RoPE and CoPE is expected to perform better for long captions. A more detailed theoretical justification or analysis could strengthen the paper by providing a solid foundation for the proposed method's expected performance improvements.

[1] Chang, Huiwen, et al. "Muse: Text-to-image generation via masked generative transformers." arXiv preprint arXiv:2301.00704 (2023).
[2] Raffel, Colin, et al. "Exploring the limits of transfer learning with a unified text-to-text transformer." Journal of machine learning research 21.140 (2020): 1-67.

**Questions:**

None

---

> ### Author Response · Authors · 2024-11-22
> **Response to Reviewer XgPN**
>
> We thank Reviewer XgPN for the constructive feedback and insightful comments. We hope to address the concerns of the Reviewer with the responses below.
>
> **Clarification of the primary contributions**. We propose the first contrastive vision-language model with relative positional encoding for long caption understanding. Our proposed approach is a substantial improvement over prior work and goes beyond a straightforward application of RoPE and CoPE. While RoPE and CoPE are established techniques, our primary contribution lies in designing a framework that can leverage any type of relative positional encodings to overcome the fundamental limitations of absolute positional encoding. Unlike Long-CLIP (ECCV 2024), which relies on absolute positional encodings combined with position stretching (fixing the first 20 tokens and interpolating positions for the rest up to 77 tokens), we propose a framework that learns pairwise relationships between any number of tokens, enabling a more generalizable long caption understanding. To further highlight the primary contribution of using relative encodings, we performed a new experiment to train our model with absolute positional encodings. We report results on long-caption retrieval tasks (top R@1) and observe significant performance drops, as detailed in the table below.
>
> | Method | Long-DCI I2T | Long-DCI T2I | ShareGPT4V I2T | ShareGPT4V T2I | Urban-1K I2T | Urban-1K T2I |
> |---------------------|--------------|--------------|----------------|----------------|--------------|--------------|
> | TULIP w/ Absolute | 41.9         | 40.0         | 96             | 93.8           | 72.9         | 69.4         |
> | **TULIP w/ Relative (RoPE)**     | **55.7**        | **56.4**         | **99.0**           | **99.0**           | **90.1**         | **91.1**         |
>
> We hope this clarifies the scope and significance of our primary contributions, which we will better stress in the introduction of the paper.
>
> **Comparison to T5-based models.** We acknowledge the strengths of T5-based text encoders in handling long text descriptions. However, T5-based models are not well-suited for vision-language tasks. In particular, T5’s text encoder is not inherently aligned with the image modality and would require considerable additional training to establish the image-text alignment - that is central to TULIP. We also note the Reviewer’s suggestion to compare with Muse, a T5-based image generation model. Unfortunately, this model is not publicly available. Therefore, we compare TULIP-based image generation with similar open-source T5-based models like PIXART-Alpha [a] and ELLA [b]. These results, included in the Appendix of the revised paper (Figures 10 and 11), demonstrate that TULIP effectively generates images with nuanced details while maintaining its cross-modal alignment without any costly retraining.
>
> [a] Chen et al., “PIXART-Alpha: Fast training of diffusion transformer for photorealistic text-to-image synthesis” (ICLR 2024)
>
> [b] Hu et al., “ELLA: Equip Diffusion Models with LLM for Enhanced Semantic Alignment” (arXiv 2024)
>
> **Theoretical analysis.** RoPE builds upon a mathematical foundation for long-context modeling by its computation of token relationships in a translation-invariant manner. This is useful for handling long contexts where absolute positional information loses relevance. In existing literature, several works attempt to provide a better theoretical understanding of RoPE mechanisms [c, d] in language models. For instance, some of them explore the frequency spectrum of RoPE, showing that lower frequencies primarily encode semantic information while higher frequencies focus on positional relationships [c]. However, a general theoretical framework has not been fully established yet. We agree with the Reviewer that a theoretical analysis is important and we will provide a discussion on the points we raised above in the revised version of the paper.
>
> [c] Barbero et al., Round and Round We Go! What makes Rotary Positional Encodings useful? (arXiv 2024)
>
> [d] Ruscio et al., Beyond position: how rotary embeddings shape representations and memory in autoregressive transformers. (arXiv 2024)
>
> We hope this addresses your concerns. We appreciate your consideration and hope this motivates you to upgrade your score. Please let us know if you have any further questions or feedback.

---

> > ### Comment · Reviewer_XgPN · 2024-11-27
> > **Response to Rebuttal**
> >
> > Dear Authors,
> >
> > Thank you for your detailed responses. While I acknowledge the efforts and appreciate the detailed clarifications provided, I still have some reservations regarding the novelty of the proposed work. Specifically, I believe the innovation in this work might not be substantial enough.
> >
> > That being said, I am willing to increase my score to a 6 if the final version of the paper includes a more comprehensive comparison with methods based on T5 or Llama text encoders, as these encoders are well-known for their capability to handle long contexts effectively.
> >
> > Thank you for your consideration, and I look forward to seeing the improvements in the revised version.

---

> ### Author Response · Authors · 2024-11-25
> **Follow-Up on Rebuttal Questions**
>
> Dear Reviewer XgPN,
>
> Thank you once again for taking the time and effort to review our submission. As the deadline for the discussion phase is approaching, we wanted to kindly check if you have any additional questions or concerns that we could address in our rebuttal.
>
> We appreciate your insights and look forward to your feedback.
>
> Best regards,
>
> The Authors

---

> ### Author Response · Authors · 2024-11-28
> **Response to Reviewer XgPN**
>
> Dear Reviewer XgPN,
>
> Thank you for your continued encouragement and guidance.
>
> Following your suggestions to compare against T5-based models, we have expanded the comparison presented in Figures 10 and 11 of the Appendix. We included additional text prompts and corresponding generated images using publicly available T5-based image generation models (PIXART-Alpha and ELLA), and SDXL + TULIP model. We provide the qualitative results in Figures 15 - 20 in the Appendix, where we can observe that our TULIP approach demonstrates overall better capability in handling nuanced descriptions. We used green image frames to highlight images that align correctly with the prompt and red frames to indicate those with missing elements. Also, note that our approach is plug-and-play for image generation, whereas the counterpart models require specific fine-tuning to perform this task. To further honor your request we will update Figure 3 in the main paper, showcasing qualitative results comparing the T5-based models to our model.
>
> In addition, given the time constraints, we performed a preliminary small-scale human evaluation to provide a quantitative comparison as well. We performed two experiments to compare to T5-based models, first to compare SDXL + TULIP to PIXART-Alpha and second to compare to ELLA. For each experiment, we gathered 5 annotators and presented 20 text prompts each with two images generated by the corresponding models. The experimental setup is shown in Figure 14 of the Appendix. The annotators are asked to assess the prompt-image alignment, by paying attention to the nuanced details of the image such as object attributes, numeracy, relationships, etc. We observe promising results as follows: compared to PIXART-Alpha we achieved a 64% win rate and compared to ELLA we achieved a 72% win rate. Currently, we are working on conducting a larger-scale human evaluation to further improve the comprehensiveness of our results. However, as the deadline for updating the paper has already passed, we will include all the details in the final version of the paper.
>
> Thank you once again for your valuable feedback. If you have any remaining questions or suggestions, we are most willing to address them.
>
> Sincerely,
>
> The Authors

---

> > ### Comment · Reviewer_XgPN · 2024-11-29
> >
> > Thank you for your detailed and thoughtful response. I appreciate the effort that you have put into addressing my suggestions and expanding the comparisons with LLM based methods, such as T5.
> >
> > The planned larger-scale human evaluation will undoubtedly add further robustness to your findings, and I look forward to seeing these results in the final version of the paper.
> >
> > Based on your thorough and well-presented rebuttal, I am willing to raise my score to 6.

---

> > > ### Author Response · Authors · 2024-11-29
> > > **Thank you for the feedback and increased score**
> > >
> > > Dear Reviewer XgPN,
> > >
> > > Thank you for acknowledging our rebuttal and efforts to address your suggestions. We greatly appreciate your thoughtful feedback and the decision to raise the score.
> > >
> > > Sincerely,
> > >
> > > The Authors

---

### Author Response · Authors · 2024-11-22
**General response by the Authors**

We sincerely thank all Reviewers for their time and constructive feedback. We appreciate that our paper is considered as a creative (xgPN) and innovative approach (3TVK), and of interest to the community (FB4m). Additionally, we appreciate the positive feedback about the paper being clear and easy to understand (FB4m) well-written and clearly presented (xgPN), and the writing is easy to follow (3TVK). We address the questions raised by the Reviewers in the respective comment sections.

---

### Meta-Review · Area_Chair_55UD · 2024-12-09

**Metareview:**

This paper presents TULIP, which aims to enhance CLIP's capability to process long captions by replacing fixed positional embeddings with relative positional encodings, extending the model's inherent context window beyond the 77 token limit. After rebuttal, it received scores of 668. All the reviewers are happy about the paper, commenting that (1) the paper is well-written and clearly presented; (2) though the proposed method is not entirely novel, the application of RoPE and CoPE to tackle the context-length limitation of CLIP is still very interesting and important; (3) results are convincing. Therefore, the AC would like to recommend acceptance of the paper.

**Additional Comments On Reviewer Discussion:**

The rebuttal was quite successful, and in fact, two of the reviewers have increased the scores after rebuttal, resulting in the all positive review feedback. Specifically,

1. Reviewers have asked more results compared with LLM-like text encoders, which may have already possess the long-context processing capability. During rebuttal, the authors have added additional results with T5-based text-to-image generation models.

2. Reviewers have asked providing more experiment details and results. During rebuttal, the authors have provided more details regarding human annotator, more qualitative comparison for short context, additional results on DOCCI, and a human evaluation on image generation results.

3. Novelty when compared with Long-CLIP. The authors have also done a nice rebuttal on this.

Overall, the rebuttal has addressed many of the reviewers' concerns.

---

### Decision · Program_Chairs · 2025-01-22

Accept (Poster)